# A predictive model using the mesoscopic architecture of the living brain to detect Alzheimer's disease

Marianna Inglese [1], Neva Patel[2], Kristofer Linton-Reid[1], Flavia Loreto[3], Zarni Win[2], Richard J. Perry[3,4], Christopher Carswell [4,5], Matthew Grech-Sollars [1,6], William R. Crum[1,7], Haonan Lu [1], Paresh A. Malhotra[3,4], the Alzheimer's Disease Neuroimaging Initiative* & Eric O. Aboagye [1✉]

**Abstract**

**Background** Alzheimer's disease, the most common cause of dementia, causes a progressive and irreversible deterioration of cognition that can sometimes be difficult to diagnose, leading to suboptimal patient care.

**Methods** We developed a predictive model that computes multi-regional statistical morpho-functional mesoscopic traits from T1-weighted MRI scans, with or without cognitive scores. For each patient, a biomarker called "Alzheimer's Predictive Vector" (ApV) was derived using a two-stage least absolute shrinkage and selection operator (LASSO).

**Results** The ApV reliably discriminates between people with (ADrp) and without (nADrp) Alzheimer's related pathologies (98% and 81% accuracy between ADrp - including the early form, mild cognitive impairment - and nADrp in internal and external hold-out test sets, respectively), without any a priori assumptions or need for neuroradiology reads. The new test is superior to standard hippocampal atrophy (26% accuracy) and cerebrospinal fluid beta amyloid measure (62% accuracy). A multiparametric analysis compared DTI-MRI derived fractional anisotropy, whose readout of neuronal loss agrees with ADrp phenotype, and *SNPrs2075650* is significantly altered in patients with ADrp-like phenotype.

**Conclusions** This new data analytic method demonstrates potential for increasing accuracy of Alzheimer diagnosis.

**Plain Language Summary**

Alzheimer's disease is the most common cause of dementia, impacting memory, thinking and behaviour. It can be challenging to diagnose Alzheimer's disease which can lead to suboptimal patient care. During the development of Alzheimer's disease the brain shrinks and the cells within it die. One method that can be used to assess brain function is magnetic resonance imaging, which uses magnetic fields and radio waves to produce images of the brain. In this study, we develop a method that uses magnetic resonance imaging data to identify differences in the brain between people with and without Alzheimer's disease, including before obvious shrinkage of the brain occurs. This method could be used to help diagnose patients with Alzheimer's Disease.

[1] Department of Surgery and Cancer, Imperial College London, London, UK. [2] Department of Nuclear Medicine, Imperial College NHS Trust, London, UK. [3] Department of Brain Sciences, Imperial College London, London, UK. [4] Department of Clinical Neurosciences, Imperial College NHS Trust, London, UK. [5] Department of Neurology, Chelsea and Westminster Hospital NHS Foundation Trust, London, UK. [6] Department of Medical Physics, Royal Surrey NHS Foundation Trust, Guilford, UK. [7] Institute for Translational Medicine and Therapeutics, Imperial College London, London, UK. *A list of authors and their affiliations appears at the end of the paper. ✉email: eric.aboagye@imperial.ac.uk

Alzheimer's disease (AD) is the most common cause of dementia worldwide and is characterised by progressive cognitive impairment and brain atrophy[1]. The disease is characterised by several events. The National Institute on Aging and Alzheimer's Association has proposed a classification system to categorise individuals based on biomarker evidence of pathology. This is called the ATN classification system and is used to rate people for the presence of cerebrospinal fluid β-amyloid (CSF Aβ or amyloid positron emission tomography (PET): 'A'), hyperphosphorylated τ (CSF pτ or τ PET: 'T'), and neurodegeneration (atrophy on structural magnetic resonance imaging (MRI), FDG) PET, or CSF total τ: 'N'), resulting in eight possible biomarker combinations[2]. Furthermore, a recent report on the involvement of microglial activation in the spread of τ tangles over the neocortex in AD suggests an additional inflammation biomarker for AD[3]. The most consistent structural imaging finding in AD is the reduced hippocampal volume[4], but this is arguably not the most specific structural biomarker as AD frequently presents with non-amnestic symptoms with initial involvement of extra-temporal regions of the brain[5]. Furthermore, the reduced hippocampal volume has been found in many other neuropsychiatric conditions including schizophrenia[6], depression[7] and hippocampal sclerosis[8] as well as the recently described limbic-predominant age-related TDP-43 encephalopathy[9]. Together with the hippocampal volume, Aβ(1–42), phosphorylated τ (pτ), and total τ (τ) CSF biomarkers have been shown to discriminate patients with AD from healthy controls[10]. However, their introduction into clinical practice is limited by considerable variability between laboratories and assay batches[10]. Similarly, blood-based biomarkers, which are eagerly awaited to address issues related to the invasiveness and high cost of CSF-based ones, often stall in the early stages because of a disconnect between academia, where biomarkers are identified, and industry, where they should be developed and commercially distributed[11].

In these last 40 years, improved computational power and storage capacity have led to numerous advances in developing non-invasive and low-cost structural biomarkers for AD that combine neuroimaging approaches, in particular structural MRI[12], with machine learning. This approach involves the acquisition of image data, the segmentation of the region of interest (ROI), feature extraction and selection for classification/prediction. Critically, features extracted from radiological images are able to reveal useful new biology[13,14] hidden to the clinician's eye[15]—at a mesoscopic scale. For example, the mesoscopic architecture of entire tumours can reveal stromal phenotype or immune context, with strong prognostic or predictive utility[16,17]. In a radiomics analysis, the extracted features represent statistical morpho-functional traits of intensity, shape, texture, scale, grey level co-occurrence matrix (GLCM), grey level run-length matrix (RLM), grey level size zone matrix (GLSZM), neighbourhood grey tone difference matrix (NGTDM) and neighbourhood grey level dependence matrix (NGLDM)[18]. A number of studies have shown texture differences between AD patients and healthy controls (HC) in structures such as the hippocampus, corpus callosum, and thalamus[19,20]. Supplementary Data 1 summarises the results and methods of the most cited papers published in the last 5 years on the classification of AD and AD-related mild cognitive impairment (MCI) patients using multimodal features. Zhang et al.[21] for instance used a single-hidden-layer neural network and predator-prey particle swarm optimisation algorithm to classify HC from AD patients. They extracted texture features from one selected axial slice of a T1-weighted (T1w) MRI scan and obtained 93% accuracy in an internal test set. Similarly, Sorensen et al.[22], with a linear discriminant analysis extracted cortical thickness measurements, volumetric measurements and hippocampal volume, shape and texture features and reached

from a T1w MRI scan with 63% accuracy. With the integration of genetic and cerebrospinal fluid biomarkers, Tong et al.[23] reached a 0.78 area under the curve (AUC) in the discrimination between HC and people with an AD-related mild cognitive impairment, thus pushing the technology towards earlier detection. They used a non-linear graph fusion method to reduce the number of volumetric features extracted from T1w MRI, intensity features extracted from PET data, three CSF measures and one genetic categorical feature. An improved performance was obtained with the view-aligned hypergraph learning approach used by Lin et al.[24]. They obtained 93, 90, 80 and 79% accuracies in the discrimination between HC and AD patients, HC and progressive MCI, HC and MCI, and stable and progressive MCI patients, respectively. In aggregate, when all patients, including control, prodromal forms of AD and AD are combined, most methods reach lower accuracy values. Of note, in most studies, models were trained and tested on an internal dataset only (Supplementary Data 1).

This current study proposes a method able to characterise early and later forms of Alzheimer's disease with the extraction from a T1w MRI sequence of 29,520 statistical morpho-functional traits distributed over a multi-regional brain mask obtained with an automatic segmentation. Healthy brain and diseases unrelated to AD pathology, including Parkinson's disease and frontotemporal dementia have been combined for the development of a set of tools able to reveal the mesoscopic architecture unique to AD.

## Methods

The study workflow is summarised in Fig. 1. The analysis of baseline age-matched T1w MRI images consisted of a two-step combined approach with and without the additional information given by cognitive scores and CSF-based biomarkers. The model was trained on 1.5 T T1w MRI scans obtained from the Alzheimer's Disease Neuroimaging Initiative (ADNI). After stratified randomisation, 70% of data were used for training and 30% for validation (robustness test shown in Supplementary Fig. 1). The control group (nADrp) included healthy controls, patients with frontotemporal dementia and with Parkinson's disease and the disease group (ADrp) included people with AD-related mild cognitive impairment (referred to as $MCI_{AD}$ in the text) and with Alzheimer's disease. The method was tested on four cohorts: (1) The unseen 1.5 T ADNI cohort (30% of the entire 1.5 T cohort, made up of 65 CN, 62 $MCI_{AD}$, 54 AD, 28 FTD and 25 PD); (2) The unseen 1.5 T dataset: 64 people obtained from the Open Access Series of Imaging Studied (OASIS) consortium with baseline T1w MRI scan and the mini-mental state examination (MMSE) score (53 CN and 11 AD); (3) The unseen 3 T dataset: 402 people obtained from ADNI with T1w MRI scan, MMSE, logical memory delayed recall total (LDELTOTAL), Aβ, τ and pτ (172 CN, 161 $MCI_{AD}$ and 69 AD); (4) The 'real-world' memory clinic cohort (IMC cohort): 83 patients with atypical presentations who underwent clinical Amyloid PET imaging as part of their diagnostic workup with a 1.5 T T1w MRI scan (45 amyloid-negative (AMY−) and 38 amyloid-positive (AMY+)) and LDELTOTAL and MMSE scores (for a subgroup of 22 people: 11 AMY− and 11 AMY+).

For the IMC cohort, we received ethical approval from the Camden and Kings Cross UK Research Ethics Committee (IRAS n. 273966) to perform retrospective anonymised and unlinked analysis of all clinical data (including MR images), provided that these were anonymised at source by a member of the clinical care team. In particular, the study protocol states: 'For all patients undergoing Amyloid PET at Imperial College Healthcare NHS Trust (ICHT) from December 2013 to January 2023 we will

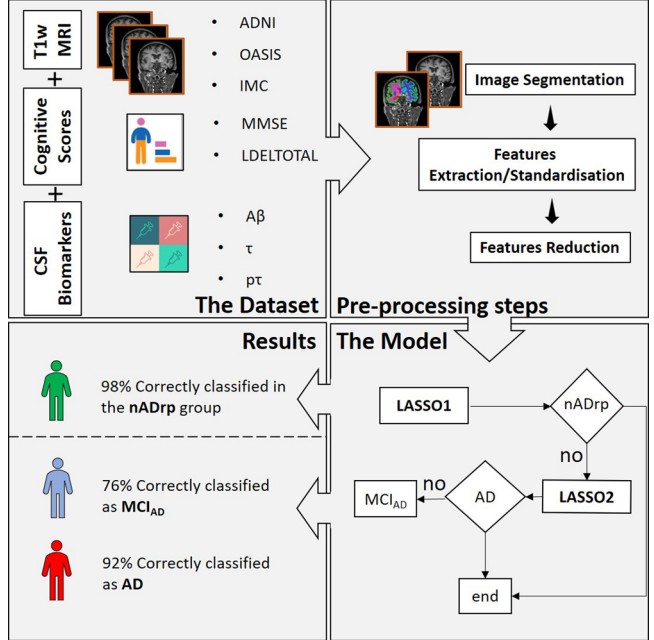

**Fig. 1 Overview of the study design and two-step least absolute shrinkage and selection operator (LASSO) approach.** Data used in this work were obtained from ADNI database, the OASIS consortium and the hospital memory clinic (IMC Cohort). Age-matched T1w MRI images were collected and segmented into 115 brain regions using the FreeSurfer's recon-all function. Isotropic (1 × 1 × 1) T1w MRI scans and their brain masks were used for the radiomic analysis in a combined double step approach. After the selection and the standardisation of features, a first least absolute shrinkage and selection operator (LASSO1) was trained to classify people into those without and with AD-related pathology (nADrp and ADrp). Within the last group, a second LASSO (LASSO2) was trained to characterise patients with a mild cognitive impairment due to AD (MCI$_{AD}$) from AD patients. The model was also integrated with cognitive scores (MMSE and LDELTOTAL) and CSF-based biomarkers (Aβ, τ and pτ). As the final algorithm was to be used to discriminate between ADrp and nADrp, combined healthy controls and patients affected by other non-AD pathologies (e.g. Frontotemporal dementia and Parkinson's disease dementia) were combined into one group referred to as non-AD-related pathology group. Initial analysis of T2w MRI data did not yield discriminatory information, so only T1w MRI data is reported.

perform retrospective anonymised and unlinked analysis of clinically collected data. This will be anonymised at source by members of the clinical care team. The data will be unlinked and there will be no prospective element to this data collection.' Informed consent was waived, as is the case for retrospective analysis of anonymised imaging data.

Data for ADNI and OASIS are openly available upon registration of investigator interest. All participants provided informed consent. Details about the Ethics statement of the ADNI study population can be found at: https://adni.loni.usc.edu. Details about the Ethics statement of the OASIS study population can be found at: https://www.oasis-brains.org/#data. Protocols for data collection and the list of institutions who approved data collection can be found at https://adni.loni.usc.edu/methods/documents/ for ADNI. OASIS is made available by the Washington University Alzheimer's Disease Research Center, the Howard Hughes Medical Institute (HHMI) at Harvard University, the Neuroinformatics Research Group (NRG) at Washington University School of Medicine, and the Biomedical Informatics Research Network (BIRN).

**MRI segmentation and radiomic analysis.** T1w MRI images were segmented to brain masks of 115 sub-regions using the FreeSurfer's *recon-all* function (45 regions obtained from the segmentation of the white matter +70 subcortical regions obtained from the additional segmentation of the cortex)[25,26]. Before segmentation, this function performs many pre-processing steps, including bias correction, image sampling and coregistration; the steps and brain regions extracted are summarised in Supplementary Table 1. The multi-regional brain masks were post-processed for the extraction of 656 features for each region using in-house software (TexLAB 2.0), which runs on MATLAB[16]. The extracted features are related to the shape and size, intensity, texture and wavelet decompositions of isotropic (1 × 1 × 1) T1w MRI scans (Supplementary Data 2). The standardised radiomic features with a false discovery rate (FDR) <5% were selected as the input for the LASSO. Tenfold cross-validation was performed to select lambda which yielded the minimum cross-validated mean squared error. The weighted sum of the selected features gave the Alzheimer's predictive Vector, ApV. For improving the model performance, the method was integrated with two cognitive measurements (MMSE and LDELTOTAL) and three CSF-based biomarkers (Aβ, τ and pτ). The result was a second predictive vector: ApV$_s$.

The model is composed of two steps:

1. In the first stage of the classification, the algorithm works on the discrimination of people with an Alzheimer related pathology. The two inputs to the LASSO1 are the nADrp group, which includes healthy controls and people with Parkinson's and frontotemporal dementia, and the ADrp group, which includes people with MCI$_{AD}$ and AD. The result of the LASSO is a reduced number of features/regions with their correspondent weights. The weighted sum of regions/features gives the ApV$_1$ (ApV$_{1s}$ with the inclusion of cognitive scores and CSF related biomarkers). People classified as not-nADrp are used as inputs for the second stage of the classification.

2. In the second stage of the classification, the algorithm works on the distinction between people with an AD-related mild cognitive impairment and with Alzheimer's disease. The LASSO2 performs a weighted sum of selected features/regions and gives the ApV$_2$ (ApV$_{2s}$ with the inclusion of cognitive scores and CSF related biomarkers) which characterise a prodromal from a late phase of AD.

The performance of the algorithm was tested using two methods. In Method A, the features extracted from the 45-region brain mask (alone and together with cognitive/CSF scores) were used and, in Method B, features extracted from the (45 + 70)-region brain mask (alone and together with cognitive/CSF scores) were used. Based on the accuracy and the accuracy/AUC values, Method B was chosen for the computation of the ApV$_1$, and Method A was chosen for the computation of ApV$_{1s}$, ApV$_2$ and ApV$_{2s}$ (Table 1).

**Genomic analysis.** Six genome-wide association study (GWAS) analyses were performed across three phenotypes (nADrp, MCI$_{AD}$, AD) derived from three variables (original label (ADNI), ApV and ApVs). One GWAS was performed for nADrp vs MCI$_{AD}$ and another GWAS for nADrp vs AD across all five variables. APOE4 allele status was provided by ADNI APOE genotype dataset. All the GWAS analyses were adjusted for age and gender using the GWASTools R package (v1.36). Each GWAS analysis calculated the main effects of all single-nucleotide polymorphisms (SNPs) on the target label (MCI$_{AD}$ /AD). For all

**Table 1 Methods comparison.**

| | | | AUC | Threshold | Specificity | Sensitivity | Accuracy | PPV | NPV |
|---|---|---|---|---|---|---|---|---|---|
| METHOD A (45 regions) | nADrp vs ADrp | T1w MRI | 0.9047 | −0.0387 | 0.8224 | 0.8362 | 0.8284 | 0.7823 | 0.8681 |
| | | T1w MRI + scores | 0.9971 | −0.1969 | 0.9671 | 0.9310 | 0.9554 | 0.9558 | 0.9484 |
| | $MCI_{AD}$ vs AD | T1w MRI | 0.7942 | 0.0648 | 1.0000 | 0.5185 | 0.7759 | 1.0000 | 0.7045 |
| | | T1w MRI + scores | 0.9656 | 0.8184 | 0.9384 | 0.8583 | 0.8633 | 0.9237 | 0.8839 |
| METHOD B(45 + 70 regions) | nADrp vs ADrp | T1w MRI | 0.9920 | 0.0938 | 0.9831 | 0.9741 | 0.9786 | 0.9826 | 0.9748 |
| | | T1w MRI + scores | 0.9859 | 0.6318 | 0.9830 | 0.9741 | 0.9786 | 0.9826 | 0.9747 |
| | $MCI_{AD}$ vs AD | T1w MRI | 0.7984 | 0.2554 | 0.9516 | 0.5556 | 0.7672 | 0.9091 | 0.7108 |
| | | T1w MRI + scores | 0.9367 | 0.1428 | 0.8871 | 0.8333 | 0.8621 | 0.8654 | 0.8594 |

The classification between nADrp and ADrp, as well as the classification between $MCI_{AD}$ and AD patients were tested with two methods.
With Method A, the algorithm received as input features extracted from the 45 brain regions resulting from segmentation of the white matter (without and with the CSF/cognitive scores). Method B considered the features extracted from the 70 subcortical regions (without and with the CSF/cognitive scores).

GWAS the empirical $p$ values were based on the Wald statistic[27]. Manhattan plots were used to visualise GWAS results.

**Statistics and reproducibility**. Standard statistical analysis was applied to all the figures as appropriate and indicated in the figure legends. All samples were used once. Multiple testing was corrected with the FDR method. All the statistical analyses were conducted in Matlab R2019b.

**Reporting summary**. Further information on research design is available in the Nature Research Reporting Summary linked to this article.

## Results

**Characteristics of data and patients**. Data used in this work were obtained from the ADNI database (www.loni.ucla.edu/ADNI), launched in 2003 as a public-private partnership, led by Principal Investigator Michael W. Weiner, MD. The primary goal of ADNI is to test whether serial MRI, PET, other biological markers, and clinical and neuropsychological assessment can be combined to measure the progression of MCI and early AD. For up-to-date information, see www.adni-info.org. From this database, all people for whom baseline MRI data (T1w magnetisation-prepared rapid acquisition with gradient echo (MP-RAGE) sequence at 1.5 T), age, and cognitive scores (MMSE[28], a brief screening test for cognitive status and the LDELTOTAL[29], a measure of verbal episodic memory), CSF-based biomarkers (Aβ, τ and pτ) were available have been included.

For the diagnostic classification at baseline, the method was trained on 783 people scanned at 1.5 T (ADNI1 cohort). They were grouped as 216 healthy controls, 208 people with MCI due to AD ($MCI_{AD}$), 181 AD, 94 patients with Frontotemporal Dementia (FTD), and 84 with Parkinson's disease (PD).

In particular, based on the data obtained from the ADNI database, two new groups of people were defined: the nADrp group, which contains people who do not show any pathology related to AD (healthy controls, PD and FTD were included here); and the ADrp group which, on the contrary, contains people with MCI due to AD and AD patients.

The method was externally tested on:

- An unseen 1.5 T dataset obtained from the OASIS consortium (https://www.oasis-brains.org/) of 64 people for whom baseline T1w sequence, age and MMSE scores were available (53 CN and 11 AD).
- An unseen 3 T dataset of 402 people obtained from the ADNI3 cohort for whom baseline T1w sequence, age, cognitive scores and CSF related biomarkers were available (172 CN, 161 $MCI_{AD}$ and 69 AD).

- The IMC cohort: 83 patients with atypical presentations who underwent clinical Amyloid PET imaging at the Imperial Memory Centre (IMC, London, UK) as part of their diagnostic workup with a 1.5 T T1w MRI scan. Of the 396 patients who had an Amyloid PET scan between December 2013 and June 2019, those ($n = 83$) who had an MRI scan available acquired between 3–6 months after the Amyloid PET scan and received a clinical neuropsychological assessment which included the administration of the Logical Memory Test, were included to the study. Of these, a subgroup of 22 patients also had an MMSE administered within 12 months of MRI scanning. At the Memory Centre, the decision to perform a clinical Amyloid PET scan is made by consensus within the Cognitive Neuroradiology Multidisciplinary Team[30] and referral to Amyloid imaging is in line with the Appropriate Use Criteria published by the Amyloid Imaging Taskforce[31]. These criteria recommend the use of clinical Amyloid PET in three main categories of patients: (1) with persistent/progressive unexplained MCI; (2) with atypical course or aetiologically mixed presentation; (3) with early age of onset. Moreover, patients undergoing clinical Amyloid PET imaging should report objective cognitive impairment with substantial diagnostic uncertainty following a comprehensive evaluation[31]. For the IMC cohort, mainly employed for the classification/evaluation of earlier diseases using structural MRI, all images were visually read as 'amyloid-positive' (AMY+, $N = 45$) or 'amyloid-negative' (AMY−, $N = 38$) by an experienced nuclear medicine radiologist using greyscale images. All AMY + patients received a clinical diagnosis of AD. AMY− patients were either diagnosed with another neurodegenerative disease (progressive non-AD MCI ($N = 4$), MCI due to hypertensive microvascular disease ($N = 1$), unspecified neurodegenerative disease (NDG) ($N = 1$), MCI due to previous stroke ($N = 1$), NDG with Parkinsonian features ($N = 1$), Lewy body dementia ($N = 1$), tauopathy ($N = 1$), normal pressure hydrocephalus ($N = 1$), isolated cerebral amyloid angiopathy ($N = 1$)) or with a non-neurodegenerative condition (e.g. depression). Patient characteristics are provided in Supplementary Fig. 2.

A multiparametric analysis was conducted on a subset of 118 diffusion tensor imaging (DTI) MRI sequences obtained from ADNI (39 AD, 40 CN and 39 $MCI_{AD}$). They were used to assess the variability of the fractional anisotropy (FA) and its relationship with the extracted features. Finally, quantitative phenotypes derived from ADNI Genetics Core were available for 199 CN, 187 $MCI_{AD}$ and 166 AD people of our 1.5 T training cohort and used for GWAS analysis.

**Radiomic predictive vector characterises Alzheimer's disease**. For each subject, T1w MRI images were automatically segmented into 115 regions from which radiomic features were independently acquired, standardised and reduced with a machine learning-based model. They were finally combined in Alzheimer's predictive vectors.

**ApV$_1$ – a biomarker to discriminate between patients with and without AD-related pathology**. Among the 656 features extracted for each of the 115 brain regions, LASSO1 selected 20 features (those with non-zero coefficients) distributed in 14 regions (Fig. 2a). The weighted sum of extracted features in the selected regions gave the Alzheimer's predictive vector ApV$_1$. With the integration of cognitive scores and CSF-based biomarkers, LASSO1 selected 19 features distributed among 12 regions (Fig. 2b). In a similar way, the combination of features, cognitive scores and regions gave the predictive vector ApV$_{1s}$. Figure 2aI, aII (and bI-bII) show the tenfold cross-validated deviance of the LASSO fit and the feature coefficients plotted against the shrinkage parameter lambda extracted for the ApV$_1$ (ApV$_{1s}$). Figure 2aIII, aIV show the ROC curve for the validation of ApV$_1$ (AUC of 0.99) and the distribution of the validated ApV$_1$ in the nADrp and ADrp groups, respectively. Similarly, Fig. 2bIII, bIV show the ROC curve for the validation of ApV$_{1s}$ (AUC of 0.99) and the distribution of the validated ApV$_1$ in the nADrp and ADrp group, respectively. The predictive ability of the ApV$_1$ in discriminating people without AD-related pathologies (nADrp) from those with AD-related pathology (ADrp) was compared to the clinical standard measures of hippocampal volume and CSF Aβ (Table 2). Of note, the measurements of diagnostic accuracy of Aβ are obtained with the application of established cut-off values[32] from the comparison between CN and ADrp. Compared to the standard measures, our method showed higher specificity, sensitivity, accuracy, negative and positive predictive values, likelihood ratios and diagnostic odds ratios. ApV$_1$ showed a state-of-the-art accuracy of 0.98 (0.26 and 0.62 for the volume of the hippocampus and CSF Aβ, respectively) in the prediction of AD-related pathologies. Of note, neither age nor CSF biomarkers were selected by LASSO1.

The testing of the method on the unseen 1.5 T OASIS cohort showed 0.81 and 0.83 accuracies for ApV$_1$ and ApV$_{1s}$, respectively (Table 2). Applied unmodified to a different field strength (3 T), our method showed 91 and 80% specificity, together with reduced accuracy of 0.49 and 0.47 for the ApV$_1$ and ApV$_{1s}$, respectively.

**ApV$_2$ — a biomarker to categorise ApV$_1$/ApV$_{1s}$ positive patients into prodromal (MCI$_{AD}$) and late (AD) groups**. The LASSO2 selected 8 features distributed in seven regions (Fig. 3a) with a dominance of the left brain. The weighted sum of the extracted features in the selected regions gave the Alzheimer's predictive vector ApV$_2$. With the integration of cognitive scores and CSF-based biomarkers, the LASSO2 selected 19 features distributed in 15 regions (Fig. 3b). The combination of features, cognitive scores and regions gave the predictive vector ApV$_{2s}$. Figures 3aI, aII (and bI-bII) show the tenfold cross-validated deviance of the LASSO2 fit and the feature coefficients plotted against the shrinkage parameter lambda extracted for the ApV$_2$ (ApV$_{2s}$). Figure 3aIII, aIV show the ROC curve for the validation of ApV$_2$ (AUC of 0.79) and the distribution of the validated ApV$_2$ in the MCI$_{AD}$ and AD groups, respectively. Similarly, Fig. 3bIII, bIV show the ROC curve for the validation of ApV$_{2s}$ (AUC of 0.95) and the distribution of the validated ApV$_2$ in the MCI$_{AD}$ and AD groups, respectively. The predictive ability of the ApV$_2$ in discriminating people with prodromal and later forms of AD in

comparison with the standard clinical measures—the volume of the hippocampus and the CSF Aβ—was quantified with the measures of diagnostic accuracies and is summarised in Table 3. ApV$_2$ reached an accuracy of 0.79 in the prediction of AD, with higher accuracy of 0.86 with the integration of clinical scores, independent of age and CSF biomarkers. The high accuracy is remarkable given the continuum of disease progression between MCI$_{AD}$ and AD. Applied to different field strengths (3 T), our method showed an accuracy of 0.62 and 0.82 for the ApV$_2$ and ApV$_{2s}$, respectively. The LASSO2 could not be tested on the OASIS cohort as it does not include any MCI$_{AD}$ people. In aggregate, our results show a predominant dysfunction in the left hemisphere[33]. This confirms the strong left-hemispheric lateralisation found in the early stages of the disease compared to weak right-hemispheric lateralisation found in advanced stages[34] (see also Supplementary Note 1 and Supplementary Fig. 3).

**Repeatability of the Alzheimer's predictive vectors**. The ApV methods were compared to the standard imaging measure (the volume of the hippocampus) and tested on a second T1w MRI scan obtained on the same day of the baseline scan used for training the model. The Bland–Altman plots are shown in Supplementary Fig. 4. Based on the reporting guidelines by Koo and Li[35], a one-way random effects, absolute agreement, single rater/measurement interclass correlation coefficient was evaluated and was 0.83, 0.89, 0.83 and 0.82 for ApV$_1$, ApV$_{1s}$, ApV$_2$ and ApV$_{2s}$, respectively. The interclass correlation coefficient for the hippocampal volume was 0.94. A boxplot of the distribution of the volumes of the hippocampus in the main groups is also shown in Supplementary Figure 4f. The robustness (non-random nature) of our ApV$_1$ and ApV$_2$ was further tested. Results are summarised in Supplementary Table 2. The measurements of diagnostic accuracy of ApV$_1$ (a) and ApV$_2$ (b) are obtained when the ApV is computed with the complete set of features extracted by the LASSO (Ftot), the four features with the highest weights (Ftest4) and all the possible permutations with three (Ftest3-p1, Ftest3-p2, Ftest3-p3, Ftest3-p4) and two features (Ftest2-p5, Ftest2-p6, Ftest2-p7, Ftest2-p8, Ftest2-p9 and Ftest2-p10) are reported. With regards to the ApV$_1$, Ftest4 showed a comparable performance compared to Ftot. Among all the permutations, Ftest3-p2 obtained the best performance involving the features extracted in the right middle temporal, rostral middle frontal and temporal pole (98% accuracy, 0.99 AUC). Regarding ApV$_2$, the best performance was obtained when the ApV was computed with only two features extracted from the left cerebral white matter (WM) and left Cerebellum WM (78% accuracy and 0.79 AUC).

**The ApV on 'real-world' data**. The model was tested on the IMC cohort, which includes people who underwent a clinical amyloid PET scan at our institution and are classified as Amyloid-positive (AMY+) or negative (AMY−). When applied to this 'real-world' cohort, no statistical difference was found between ApV$_1$ and ApV$_2$ in people with positive/negative amyloid enhancement ($p = 0.88$) (Supplementary Fig. 5b). Regardless of the PET output, people were classified as nADrp and MCI$_{AD}$ (in particular, of the 44 AMY−, 42 were classified as nADrp, 2 as MCI$_{AD}$ and 1 as AD; of the 38 AMY−, 36 were classified as nADrp and 2 as MCI$_{AD}$). The model was also tested on a subgroup of 22 people whose T1w MRI scan was obtained 5 ± 4 months after Amyloid PET imaging and was used together with the MMSE and the LDEL-TOTAL cognitive scores. In this small cohort, people with a negative PET scan were classified as nADrp ($N = 8$), MCI$_{AD}$ ($N = 2$) and AD ($N = 1$). People with a positive scan were evenly classified as nADrp and MCI$_{AD}$ ($N = 5$), only one subject was classified as AD. In relation to the PET output, our ApV$_{1s}$ showed

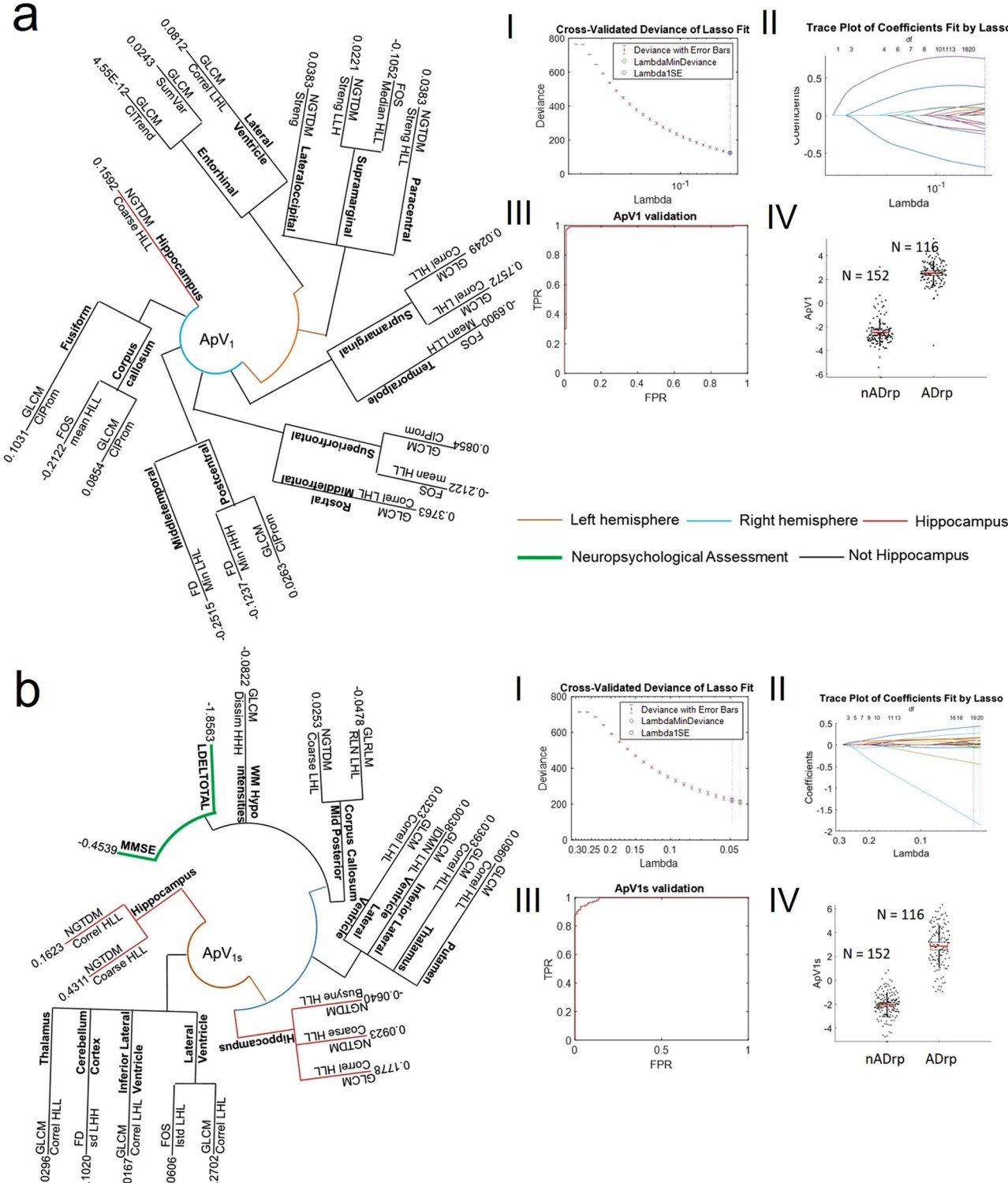

**Fig. 2 Results of LASSO1.** The biophysical mesoscopic properties of brain regions in nADrp and ADrp people are depicted by the combination of features/regions selected by the LASSO1. In the radial phylogeny trees, the components of ApV₁ (**a**), ApV₁ₛ (**b**) are summarised. **aI** and **aII** show the tenfold cross-validated deviance of the LASSO1 fit and feature coefficients plotted against the shrinkage parameter Lambda. Shown in **aIII** the ROC curve for the validation of ApV₁. Shown in **aIV** is the distribution of the validated ApV₁ in the nADrp (N = 152) and ADrp (N = 116) groups. **bI** and **bII** show the tenfold cross-validated deviance of the LASSO1 fit with the integration of cognitive scores and CSF-based biomarkers and the feature coefficients plotted against the shrinkage parameter Lambda. **bIII** and **bIV** show the ROC curve for the validation of ApV₁ₛ, and the distribution of the validated ApV₁ₛ in the nADrp (N = 152) and ADrp (N = 116) groups. In the radial trees, branches are coloured based on the region selected (hippocampus: red, other: black), their brain hemisphere (left: orange, right: blue), and the cognitive score (green). In the box plots, points are laid over a 1.96 standard error of the mean (95% confidence interval) and one standard deviation (black vertical line).

**Table 2 Diagnostic performance of the Alzheimer's predictive vector ApV₁ and ApV₁ₛ.**

| | Training 1.5 T ADNI dataset | | Unseen 1.5 T ADNI dataset | | Unseen 1.5 T OASIS dataset | | Unseen 3 T ADNI dataset | | Volume of hippocampus | Aβ |
|---|---|---|---|---|---|---|---|---|---|---|
| | ApV₁ | ApV₁ₛ | ApV₁ | ApV₁ₛ | ApV₁ | ApV₁ₛ | ApV₁ | ApV₁ₛ | | |
| AUC | 0.9981 | 0.9971 | 0.9786 | 0.9490 | 0.6706 | 0.6801 | 0.6533 | 0.5192 | 0.7790 | 0.5045 |
| Threshold | 0.0938 | −0.1969 | 0.0938 | −0.1969 | 0.0938 | −0.1969 | 0.0938 | −0.1969 | −0.1132 | 192 |
| Specificity | 0.9818 | 0.9669 | 0.9831 | 0.9671 | 0.8868 | 0.9057 | 0.9127 | 0.8081 | 0.2273 | 0.0091 |
| Sensitivity | 0.9819 | 0.9780 | 0.9741 | 0.9310 | 0.4545 | 0.4545 | 0.1739 | 0.2304 | 0.2941 | 1 |
| Accuracy | 0.9836 | 0.9728 | 0.9786 | 0.9554 | 0.8125 | 0.8281 | 0.4900 | 0.4776 | 0.2626 | 0.6236 |
| NPV | 0.9855 | 0.9750 | 0.9748 | 0.9484 | 0.8868 | 0.8889 | 0.4524 | 0.4398 | 0.2227 | 1 |
| PPV | 0.9818 | 0.9709 | 0.9826 | 0.9558 | 0.4545 | 0.500 | 0.7272 | 0.6162 | 0.2996 | 0.6223 |
| LR + | 54.4364 | 29.5952 | 57.4741 | 28.3034 | 4.0151 | 4.8182 | 1.9942 | 1.2010 | 0.3806 | 1.0009 |
| LR − | 0.0149 | 0.0868 | 0.0263 | 0.0713 | 0.6151 | 0.6023 | 0.9050 | 0.9522 | 3.1059 | 0 |
| Yi | 0.9672 | 0.9450 | 0.9572 | 0.8981 | 0.3413 | 0.3602 | 0.0867 | 0.0385 | −0.4786 | 0.0092 |
| DOR | 3653.4 | 1301.6 | 2184.6 | 396.9 | 6.5278 | 8 | 2.2035 | 1.2612 | 0.1225 | NA |

Diagnostic performance of ApV₁ and ApV₁ₛ was evaluated in the 1.5 T training dataset (ADNI), the unseen 1.5 T ADNI, 1.5 T OASIS and 3 T ADNI datasets. The performance of the ApV is also compared to the current clinically used measure of hippocampal volume in the discrimination between nADrp and ADrp patients, and CSF Aβ in the discrimination between CN and ADrp.
In the testing test, AUC values were generated from sensitivity and specificity[62].
DOR diagnostic odds ratio, Yi Youden index value, LR+ positive likelihood ratio, LR− negative likelihood ratio, NA undefined values derived from the division by zero, NPV negative predictive value, PPV positive predictive value.

a statistical difference between AMY− and AMY+ ($p = 0.02$) (Supplementary Fig. 5b).

**Genome-wide association study and fractional anisotropy.** Figure 4 shows the Manhattan plot of the GWAS for the ApVs. The Manhattan plot shows one SNP above a significance threshold of $p < 10^{-7}$. This SNP corresponded to the genotype RS IDs: *rs2075650*. The *rs2075650* SNP was above the significance thresholds across all variables, original labels, ApV and ApVs (Supplementary Figs. 6, 7). Similarly, for all cognitively normal vs mild cognitive impairment, no SNPs were above the threshold. Additionally, in the ApV group, ADrp vs AD, the $p < 10^{-6}$ SNP *rs575606* was above a threshold of $p < 10^{-6}$ (Supplementary Fig. 6). When performing a GWAS adjusting for the presence of one or two *APOE4* alleles, no SNPs were identified as significantly associated with AD in any of the outcomes (Supplementary Fig. 7). Additionally, we present LocusZoom plots of the 2000 base pairs around *rs2075650* on the GWAS results without the adjustment of APOE4 (Supplementary Fig. 8). An extensive interpretation of the GWAS results is included in Supplementary Note 2. In aggregate, Supplementary Note 2 includes the allele frequencies evaluation (allele proportions and Hardy–Weinberg Equilibrium Fisher's exact test $p$ value) for the SNP *rs2075650*, which shows 'B' to be the minor allele with both the ApVs and ApV classification (Supplementary Table 3).

In agreement with the ADrp phenotype, the analysis of fractional anisotropy from DTI MRI sequences showed a neuronal loss in ADrp people. The variation of FA was tested in 115 brain regions. A Wilcoxon rank-sum test was used to test the regional statistical difference of FA between nADrp ($N = 79$) and ADrp ($N = 39$) and between MCI$_{AD}$ ($N = 31$) and AD ($N = 8$) people. For most regions, no statistically significant reduction was present ($p > 0.05$) (Fig. 4C). Twenty-two out of 115 regions showed a significant variation of FA between nADrp and ADrp (left and right cerebral cortex and the left caudate showed an FA increase). Between MCI$_{AD}$ and AD, 11 out of 115 regions showed a significant variation of FA (an increase of FA was present only in the left amygdala). Figure 4D shows the absolute values of FA in the regions for which a statistical difference was found between nADrp and ADrp and between MCI$_{AD}$ and AD patients ($p < 0.05$).

## Discussion

This study presents a novel MRI-based radiomic predictive vector which outperforms standard hippocampal volume and CSF Aβ measurements (Table 2) reaching a 0.98 accuracy in an internal test set (mean value 0.9830, 95% confidence interval (CI) [0.9829, 0.9831]) for the triage of people without an AD-related pathology. Our ApV is robust and repeatable across MRI scans (Supplementary Fig. 4), demonstrating its potential for applicability in clinical practice in the future.

This method does not require a subject matter expert, but rather uses established software for both brain segmentation (FreeSurfer)[25,26,36] and radiomics analysis[16]. The algorithm computes manually engineered features allowing an easy interpretation of the ApV and facilitating clinical translation. To avoid overfitting, the dimensionality of the model is reduced with the 'least absolute shrinkage and selection operator'[37], which selects the most informative and less redundant features corresponding to specific brain regions. The LASSO is suitable for the regression of high-dimensional features in a radiomics strategy[38] allowing, in a single regression model, the statistical analysis of complex data where data are labelled to exploit dependence patterns in specific brain regions. Compared to the most common multivariate models present in the literature (Random Forest, Naïve

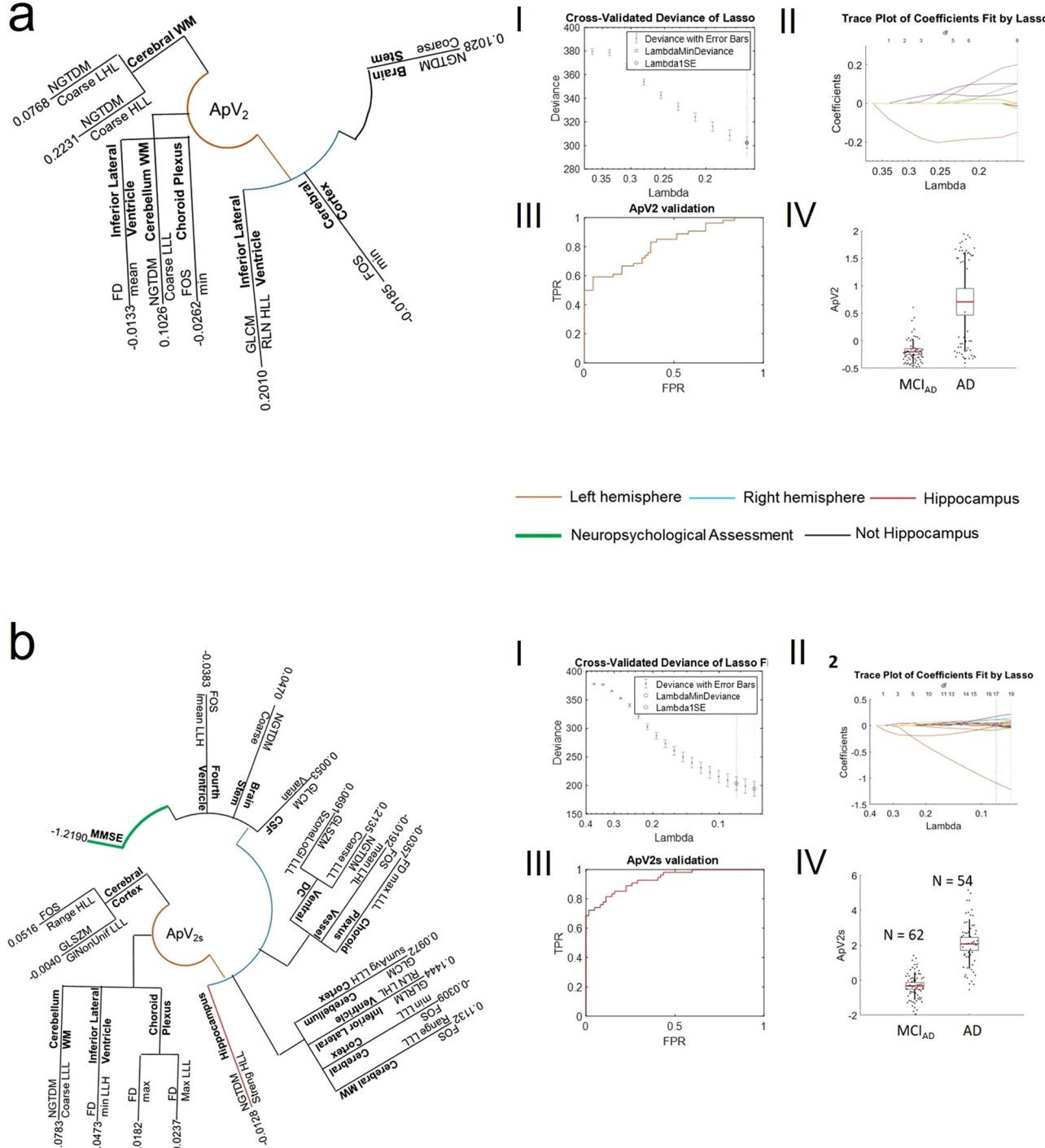

**Fig. 3 Results of LASSO2.** The biophysical mesoscopic properties of brain regions in MCI$_{AD}$ and AD people are depicted by the combination of features/ regions selected by the LASSO2. In the radial phylogeny trees, the components of ApV$_2$ (**a**), ApV$_{2s}$ (**b**) are summarised. **aI** and **aII** show the tenfold cross-validated deviance of the LASSO2 fit and the feature coefficients plotted against the shrinkage parameter Lambda. Shown in **aIII** the ROC curve for the validation of ApV$_2$. Shown in **aIV** the distribution of the validated ApV$_2$ in the MCI$_{AD}$ ($N = 62$) and AD ($N = 54$) groups. **bI** and **bII** show the tenfold cross-validated deviance of the LASSO2 fit with the integration of cognitive scores, CSF-based biomarkers and the feature coefficients plotted against the shrinkage parameter lambda. **bIII** and **bIV** show the ROC curve for the validation of ApV$_{2s}$, and the distribution of the validated ApV$_{2s}$ in the MCI$_{AD}$ ($N = 62$) and AD ($N = 54$) groups. In the radial trees, branches are coloured based on the region selected (hippocampus: red, other: black), their brain hemisphere (left: orange, right: blue), and the cognitive score (green). In the box plots, points are laid over a 1.96 standard error of the mean (95% confidence interval) and one standard deviation (black vertical line).

**Table 3 Diagnostic performance of the Alzheimer's predictive vectors ApV$_2$ and ApV$_{2s}$.**

| | Training 1.5 T ADNI dataset | | Unseen 1.5 T ADNI dataset | | Unseen 3 T ADNI dataset | | | |
| --- | --- | --- | --- | --- | --- | --- | --- | --- |
| | ApV$_2$ | ApV$_{2s}$ | ApV$_2$ | ApV$_{2s}$ | ApV$_2$ | ApV$_{2s}$ | Volume of hippocampus | Aβ |
| AUC | 0.8580 | 0.9656 | 0.7258 | 0.8983 | 0.5072 | 0.7111 | 0.5345 | 0.5 |
| Threshold | 0.3017 | 0.8184 | 0.3017 | 0.8184 | 0.3017 | 0.8184 | −0.7827 | 192 |
| Specificity | 0.9863 | 0.9384 | 0.9516 | 0.9384 | 1 | 0.9875 | 0.3387 | 0 |
| Sensitivity | 0.5590 | 0.8583 | 0.5000 | 0.8583 | 0.0289 | 0.4347 | 0.7593 | 1 |
| Accuracy | 0.7875 | 0.9011 | 0.7863 | 0.8633 | 0.6296 | 0.8217 | 0.5345 | 0.4887 |
| NPV | 0.7200 | 0.8839 | 0.6860 | 0.8839 | 0.7061 | 0.8030 | 0.6176 | NA |
| PPV | 0.9726 | 0.9237 | 0.9000 | 0.9237 | 1 | 0.9375 | 0.5000 | 0.4887 |
| LR+ | 40.8110 | 13.9230 | 10.3333 | 13.9230 | NA | 35.0000 | 1.1481 | 1 |
| LR− | 0.4471 | 0.1510 | 0.5254 | 0.1510 | 0.9710 | 0.5723 | 0.7108 | NA |
| Yi | 0.5454 | 0.7966 | 0.4516 | 0.7966 | 0.0289 | 0.4223 | 0.0980 | 0 |
| DOR | 91.2857 | 92.1790 | 19.6667 | 92.1790 | NA | 61.1538 | 1.6154 | NA |

Diagnostic performance of ApV2 and ApV2s evaluated in the 1.5 T training dataset (ADNI), the unseen 1.5 T ADNI and 3 T ADNI datasets compared to the volume of the hippocampus and Aβ in the discrimination between MCIAD and AD patients. *Of note, the measurements of diagnostic accuracy of Aβ are obtained with the application of the established cut-off values (Shaw et al.).
In the testing test, AUC values were generated from sensitivity and specificity[62].
*DOR* diagnostic odds ratio, *Yi* Youden index value, *LR+* positive likelihood ratio, *LR−* negative likelihood ratio, *NA* undefined values derived from the division by zero, *NPV* negative predictive value, *PPV* positive predictive.

Bayes, K-Nearest Neighbours and Support Vector Machine), our univariate analysis shows higher accuracy (Supplementary Table 4) and easier interpretability, thanks to the implementation of manually engineered features, facilitating clinical translation. In order to improve the model's generalisability, the training of ApV exploits commonalities and differences within the segmentations between controls and patients with FTD, PD, MCI due to Alzheimer's disease and AD—appreciating that patients who come to the memory clinic may have other conditions. We rationalised that the extra information from FTD and PD segments will allow the model to gain a better contextual understanding of the regions of interest and better discriminate nADrp from ADrp rather than for detecting FTD or PD *per se*. Appreciating that the inclusion of non-AD pathologies in the control group of the training set could have introduced a classification bias leading to an overrated model accuracy, further tests were done to assess the impact of PD and FTD patients in the nADrp group. The measurements of diagnostic accuracy obtained when the classification is computed between CN and ADrp, as well as between CN and MCI$_{AD}$ and CN and AD patients (in comparison with the proposed original method, in italic – Table 4) prove that the performance of our method is not influenced by the presence of PD and FTD patients in the nADrp group.

In an internal test set (the 1.5 T ADNI cohort), the ApV$_1$ is able to discriminate between people with (ADrp) and without (nADrp) Alzheimer's related pathologies with a 0.98 accuracy. Differently from the majority of published research studies, where models are usually trained between two categories (e.g. HC vs AD or MCI vs AD) (Supplementary Data 1), our algorithm includes both AD patients and people with the early form of AD, mild cognitive impairment in the ADrp group. This procedure permits triage of patients who neither have MCI$_{AD}$ nor AD, taking into account the notion that Alzheimer's disease exists along a spectrum, from early memory changes to functional dependence and death. To the best of our knowledge, the accuracy reached by the ApV in the internal dataset (obtained by analysing MRI data with or without cognitive scores) is superior to the ones obtained from published research studies, which focus on a single internal test set only[39–41]. However, the true performance of a radiomic model needs to be validated on external datasets or independent institutional cohorts; in practice, only a minority of studies report an application of algorithms to external datasets[42]. When tested on an external test set (the unseen 1.5 T OASIS cohort), our algorithm reaches a 0.86 accuracy, higher than previously reported studies[43]. Furthermore, when compared to the standard clinical measures of hippocampal atrophy and cerebrospinal fluid beta-amyloid concentration, the ApV shows higher accuracy, presenting a potentially valid alternative to the invasive CSF measurements.

To be precise, the ApV is independent of the amyloid levels in the CSF. Regardless of the stronger pathological biomarker signature encountered when increased CSF concentrations of τ and pτ species, decreased concentrations of Aβ[32,44] and cognitive scores are considered together with structural data, it is notable that Aβ, τ and pτ were not selected as part of the optimised ApV algorithm. This result can be explained by the inner low accuracy of the CSF-based biomarkers collected for our cohorts (Supplementary Table 5), with respect to the established cut-off values (93 pg/ml for τ, 192 pg/ml for Aβ1–42 and 23 pg/ml for pτ)[32]. The non-overlapping nature of the ApV means that a combination of these with CSF biomarkers could be explored in the future to further improve accuracy in early MCI$_{AD}$ /AD.

The ApV describes the mesoscopic architecture and the biological changes of an AD brain. With an unsupervised approach, and appreciating the lack of post-mortem AD confirmation in our cohort of people, the algorithm selects texture and shape features, strong biomarkers of AD[20,45,46], in regions typically involved in the development of the disease (the hippocampus, entorhinal cortex, amygdala[47]). In particular, our results show a predominant dysfunction in the left hemisphere[33], confirming the strong left-hemispheric lateralisation found in the early stages of the disease compared to weak right-hemispheric lateralisation found in advanced stages[34]. As extensively described in the 'Biological interpretation of ApV' in the Supplementary Note 1, the cortical grey matter structural changes, usually due to the ageing brain and cognitive decline caused by neuronal loss[48–50], are represented in part within the ApV by GLCM and FD features[51] and confirmed, with the multiparametric analysis of DTI MRI images, by the statistically significant decrease of FA in AD patients. For example, the GLCM correlation feature, filtered with an LHL wavelet filter, in the left lateral ventricle expresses the dependency of grey level values to their respective voxels in the GLCM possibly relating to grey levels' distribution in this brain region of AD patients where ventriculomegaly is commonly observed. Brain parenchymal shrinkage causes, in most neurodegenerative disorders, the passive enlargement of the lateral, third and fourth ventricles with a significant ventricular enlargement associated with AD[52]. Furthermore, cognitive decline, expressed as local neuronal loss of many hippocampal subfields

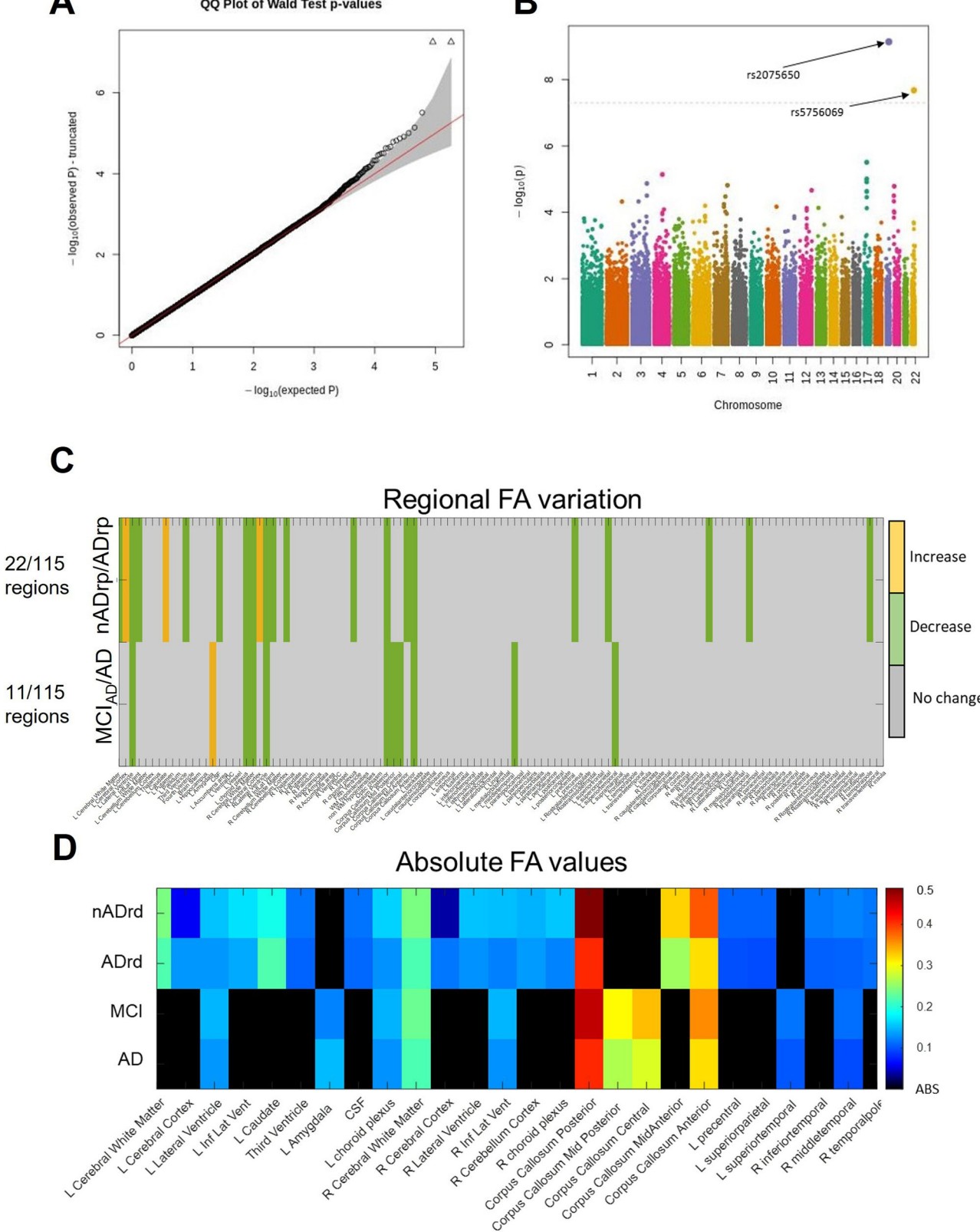

(subiculum, cornu ammonis) following AD progression (as also confirmed by the statistically significant decrease of fractional anisotropy), is expressed by the Neighbouring Grey Tone Difference Matrix (NGTDM) coarseness feature extracted in the right hippocampus. This is a measure of the average difference between the central voxel and its neighbourhood and is an indication of the spatial rate of change. A higher value indicates a lower spatial change rate and a locally more uniform texture.

**Fig. 4 Genetic and molecular characteristics associated with the ApV biomarker.** In **A**, **B** the Q–Q and Manhattan plots of genome-wide association study (GWAS) of the cognitively normal and Alzheimer's disease labels derived from ApVs are shown. In detail, **B** is the Manhattan plot of the $p$ values ($-\log_{10}$(Wald $p$ value)) from GWAS analysis of the ApVs. The horizontal line displays the cut-off for two significant levels ($p < 10^{-7}$). Shown in **A** is the quantile–quantile (Q–Q) plot of the distribution of the observed $p$ values ($-\log_{10}$(observed $p$ value)) in this sample versus the expected $p$ values ($-\log_{10}$(expected $p$ value)) under the null hypothesis of no association. Shown in **C** is the variation of fractional anisotropy tested in 115 brain regions. A Wilcoxon rank-sum test was used to test the regional statistical difference of FA between nADrp ($N = 79$) and ADrp ($N = 39$) and between $MCI_{AD}$ ($N = 31$) and AD ($N = 8$) people. **D** The absolute values of FA in the regions for which a statistical difference was found between nADrp and ADrp and between $MCI_{AD}$ and AD patients ($p < 0.05$) is shown.

---

**Table 4 Test on the diagnostic performance of the algorithm.**

|  |  | AUC | Threshold | Specificity | Sensitivity | Accuracy | PPV | NPV |
|---|---|---|---|---|---|---|---|---|
| nADrp vs ADrp | train | 0.9981 | 0.0938 | 0.9819 | 0.9853 | 0.9836 | 0.9818 | 0.9855 |
|  | test | 0.9920 | 0.0938 | 0.9831 | 0.9741 | 0.9786 | 0.9826 | 0.9748 |
| CN vs ADrp | train | 1.0000 | −0.1109 | 0.9934 | 1.0000 | 0.9976 | 0.9964 | 1.0000 |
|  | test | 1.0000 | −0.1109 | 1.0000 | 0.9828 | 0.9890 | 1.0000 | 0.9701 |
| CN vs $MCI_{AD}$ | train | 1.0000 | 0.0722 | 1.0000 | 0.9932 | 0.9966 | 1.0000 | 0.9934 |
|  | test | 1.0000 | 0.0722 | 1.0000 | 0.9839 | 0.9921 | 1.0000 | 0.9848 |
| CN vs AD | train | 0.9999 | −0.1109 | 0.9934 | 1.0000 | 0.9964 | 0.9922 | 1.0000 |
|  | test | 1.0000 | −0.1109 | 1.0000 | 0.9815 | 0.9916 | 1.0000 | 0.9848 |

The two inputs to the LASSO1 are the nADrp group, which includes healthy controls and people with Parkinson's and frontotemporal disease, and the ADrp group, which includes people with $MCI_{AD}$ and AD. The diagnostic performance of the algorithm was tested when the classification is computed between the ADrp group and healthy people, between CN and $MCI_{AD}$ and CN and AD patients.

---

Together with high pass wavelet filters applied in one dimension and a low pass one applied in the other two, the extraction of the coarseness in the hippocampus represents an index of heterogeneity. Interestingly, the algorithm also selects regions not commonly related to AD, such as the cerebellum and the ventral diencephalon. Together with a few studies reported in the literature[53,54], this outcome challenges the traditional view that white matter bundles in the cerebellum or in the ventral diencephalon are not affected by AD, possibly highlighting new therapeutic opportunities.

The GWAS performed across nADrp, $MCI_{AD}$ and AD derived from the ApV classification labels highlights genetic insights distinct from classical *APOE*-only gene association in AD. The non-causal significant alteration of the SNP *rs2075650* found in patients with ADrp-like phenotype reinforces a body of research that associates this gene with $MCI_{AD}$ and AD[55–57]. *TOM40* is located adjacent to *APOL*, and the two genes are thought to be correlated with Alzheimer's due to linkage. Given that after adjusting for *APOE4* allele status, *rs2075650* is no longer significant, this suggests the *TOM40* association signal is driven by the *APOE4* allele and surrounding variants.

The ApV is also age-independent for the age range used. The similarity between age-related atrophy in AD and in normal aging represents one limitation of applying multivariate models to structural MRI[58]. In this study, this issue is assessed following the age-correction method by Moradi et al.[59], which introduced a large distortion on the MRI image, limiting the reliability of the extracted features, thus, considering age as an additional feature. The result was a non-selection of age among the less redundant, most significant features.

This method provides a biomarker able to detect an early stage of AD with a significant potential improvement of the clinical decision support system. The ApV was tested on a clinical cohort of people with objective cognitive impairment and uncertain underlying aetiology caused by an atypical clinical course or the presence of multiple co-morbidities (Fig. 5a). When employed in this cohort, the ApV outperformed the hippocampal volume measurements (Fig. 5b) and the standard cognitive scores (Fig. 5e) showing a statistically significant difference between the AMY− and AMY+ groups

($p = 0.02$, Fig. 5d). Therefore, where isolated hippocampal atrophy or episodic memory impairment fails to differentiate AMY+ from AMY− patients, the ApV shows a stronger diagnostic potential.

Other than its retrospective nature, a limitation of this study is represented by the lower performance of the method when tested unmodified at higher different field strengths (the unseen 3 T dataset). As shown in Table 2, very high positive predictive values are associated with low sensitivity and overall low accuracy for both the $ApV_1$ and $ApV_2$ obtained from a baseline 3 T ADNI cohort. This result confirms the hypothesis that MRI radiomic features are susceptible to magnetic field strength[60] and limits the applicability of our current method only to 1.5 T data. Future studies will focus on the development of preprocessing techniques for the improvement of the performance of the algorithm on 3 T data together with the introduction of an equivalent algorithm for higher field strengths. A second limitation of this study is the impossibility of directly comparing our method with the published literature. This is mainly related to how we decided to structure our input to improve the model's generalisability: the control group, together with healthy people, also contains people with Parkinson's disease and frontotemporal dementia. A third limitation of this study is related to the computational effort needed to pre-process the structural MRI data. The segmentation step performed by FreeSurfer's recon-all function usually requires about 10/12 h per subject. In this regard, to reduce computation time, we decided to re-run the analyses in parallel using 12 logical cores: a group of 10/15 scans were segmented with this latter approach in the same amount of time. In fact, we believe that with the implementation of a faster segmentation pipeline, this work would outperform the clinical tests now used in isolation. A possible future solution to minimise segmentation time in clinical practice could be the extraction of a custom T1w-MRI-based template built from the chosen dataset (e.g. using the SPM DARTEL pipeline).

In summary, this study proposes an unsupervised approach for the development of an MRI-based biomarker for the biological characterisation of AD. The ApV is reproducible and robust. It can be easily computed with the calculation of

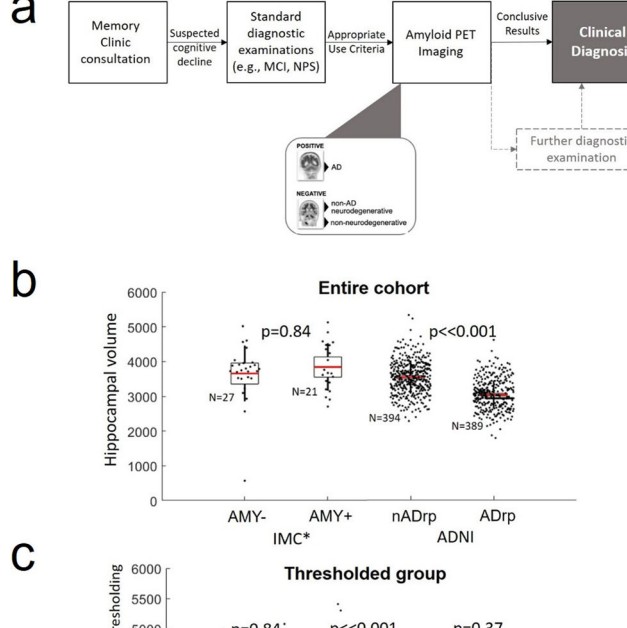

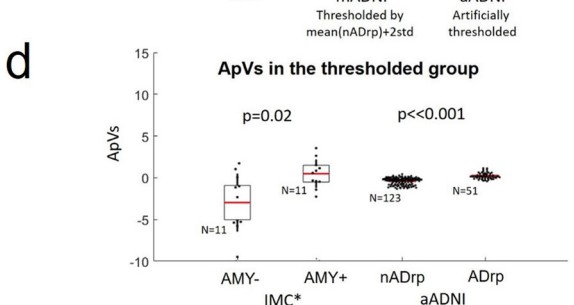

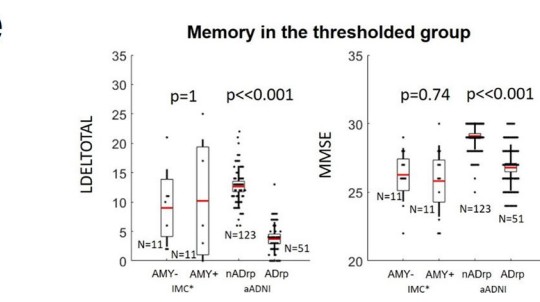

**Fig. 5 Early detection of Alzheimer's disease in an atypical-AD cohort.**
**a** Patients presenting at the IMC with suspected cognitive decline undergo a range of standard diagnostic investigations, such as MRI and neuropsychological assessment, which can vary across individuals depending on the clinical presentation. Where diagnostic uncertainty persists, the decision to perform Amyloid PET Imaging is made by consensus by a multidisciplinary team[30] and in line with the appropriate use criteria[31]. In this context, a positive Amyloid PET imaging is highly suggestive of an underlying AD diagnosis, while a negative scan rules out AD. Patients with a negative Amyloid PET imaging often have either a non-AD type of dementia (e.g., FTD) or other non-neurodegenerative causes of cognitive impairment (e.g. depression). **b** The hippocampal volumes evaluated in the entire ADNI and IMC cohorts ($N = 27$ AMY−, $N = 21$ AMY+, $N = 394$ nADrp, $N = 389$ ADrp). **c** The distribution of the hippocampal volumes in the IMC cohort and in artificially thresholded subgroups of ADNI people ($N = 27$ AMY−, $N = 21$ AMY+, $N = 387$ nADrp and $N = 340$ ADrp in the mADNI group, $N = 123$ nADrp and $N = 51$ ADrp in the aADNI group). **d** The ApVs values of the IMC* and aADNI cohorts (where the volume of the hippocampus is statistically significant between the control and disease group ($p = 0.02$)) ($N = 11$ AMY−, $N = 11$ AMY+, $N = 123$ nADrp, $N = 51$ ADrp). **e** The distribution of the LDELTOTAL and MMSE scores in the IMC* and aADNI cohorts ($N = 11$ AMY−, $N = 11$ AMY+, $N = 123$ nADrp, $N = 51$ ADrp). In the box plots, points are laid over a 1.96 standard error of the mean (95% confidence interval) and one standard deviation (black vertical line).

## Code availability

The MATLAB scripts used to reproduce the key findings and generate figures are publicly accessible in Mendeley Data with the identifier https://doi.org/10.17632/rpztyz22df.1[61].

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

manually engineered features and is ready to be integrated into the clinical decision support system without the need for additional sampling or patient testing.

## Data availability

The radiomics data generated in this study have been deposited into the Mendeley database under the accession code DOI: 10.17632/rpztyz22df[61]. All the other data supporting the findings of this study, together with the source data underlying the graphs and charts shown in the manuscript are available and have been deposited into the Mendeley database under the accession code https://doi.org/10.17632/rpztyz22df[61].

12. Hua, X. et al. Mapping Alzheimer's disease progression in 1309 MRI scans: Power estimates for different inter-scan intervals. *NeuroImage* **51**, 63–75 (2010).

13. Lambin, P. et al. Radiomics: extracting more information from medical images using advanced feature analysis. *Eur. J. Cancer* **48**, 441–446 (2012).

14. Aerts, H. J. et al. Decoding tumour phenotype by noninvasive imaging using a quantitative radiomics approach. *Nat. Commun.* **5**, 1–9 (2014).

15. Parekh, V. & Jacobs, M. A. Radiomics: a new application from established techniques. *Expert Rev. Precis. Med. Drug Dev.* **1**, 207–226 (2016).

16. Lu, H. et al. A mathematical-descriptor of tumor-mesoscopic-structure from computed-tomography images annotates prognostic-and molecular-phenotypes of epithelial ovarian cancer. *Nat. Commun.* **10**, 1–11 (2019).

17. Sun, R. et al. A radiomics approach to assess tumour-infiltrating CD8 cells and response to anti-PD-1 or anti-PD-L1 immunotherapy: an imaging biomarker, retrospective multicohort study. *Lancet Oncol.* **19**, 1180–1191 (2018).

18. Zwanenburg, A. et al. The image biomarker standardization initiative: standardized quantitative radiomics for high-throughput image-based phenotyping. *Radiology* **295**, 328–338 (2020).

19. Sørensen, L. et al. Early detection of Alzheimer's disease using M RI hippocampal texture. *Hum. Brain Mapp.* **37**, 1148–1161 (2016).

20. De Oliveira, M. et al. MR imaging texture analysis of the corpus callosum and thalamus in amnestic mild cognitive impairment and mild Alzheimer disease. *Am. J. Neuroradiol.* **32**, 60–66 (2011).

21. Zhang, Y. et al. Multivariate approach for Alzheimer's disease detection using stationary wavelet entropy and predator-prey particle swarm optimization. *J. Alzheimer's Dis.* **65**, 855–869 (2018).

22. Sorensen, L. et al. Differential diagnosis of mild cognitive impairment and Alzheimer's disease using structural MRI cortical thickness, hippocampal shape, hippocampal texture, and volumetry. *NeuroImage Clin.* **13**, 470–482 (2017).

23. Tong, T. et al. Multi-modal classification of Alzheimer's disease using nonlinear graph fusion. *Pattern Recognit.* **63**, 171–181 (2017).

24. Liu, M., Zhang, J., Yap, P.-T. & Shen, D. View-aligned hypergraph learning for Alzheimer's disease diagnosis with incomplete multi-modality data. *Med. Image Anal.* **36**, 123–134 (2017).

25. Fischl, B. et al. Whole brain segmentation: automated labeling of neuroanatomical structures in the human brain. *Neuron* **33**, 341–355 (2002).

26. Fischl, B. et al. Automatically parcellating the human cerebral cortex. *Cereb. Cortex* **14**, 11–22 (2004).

27. Purcell, S. et al. PLINK: a tool set for whole-genome association and population-based linkage analyses. *Am J. Hum. Genet.* **81**, 559–575 (2007).

28. Folstein, M. F., Folstein, S. E. & McHugh, P. R. "Mini-mental state". A practical method for grading the cognitive state of patients for the clinician. *J. Psychiatr. Res.* **12**, 189–198 (1975).

29. Wechsler, D. *WMS-R: Wechsler Memory Scale--Revised: Manual* (Psychological Corp., 1987).

30. Kolanko, M. A. et al. Amyloid PET imaging in clinical practice. *Pract. Neurol.* **20**, 451–462 (2020).

31. Johnson, K. A. et al. Appropriate use criteria for amyloid PET: a report of the Amyloid Imaging Task Force, the Society of Nuclear Medicine and Molecular Imaging, and the Alzheimer's Association. *Alzheimers Dement.* **9**, 1–16 (2013).

32. Shaw, L. M. et al. Cerebrospinal fluid biomarker signature in Alzheimer's disease neuroimaging initiative subjects. *Ann. Neurol.* **65**, 403–413 (2009).

33. Loewenstein, D. A. et al. Predominant left hemisphere metabolic dysfunction in dementia. *Arch. Neurol.* **46**, 146–152 (1989).

34. Weise, C. M. et al. Left lateralized cerebral glucose metabolism declines in amyloid-β positive persons with mild cognitive impairment. *NeuroImage Clin.* **20**, 286–296 (2018).

35. Koo, T. K. & Li, M. Y. A guideline of selecting and reporting intraclass correlation coefficients for reliability research. *J. Chiropr. Med.* **15**, 155–163 (2016).

36. Han, X. et al. Reliability of MRI-derived measurements of human cerebral cortical thickness: the effects of field strength, scanner upgrade and manufacturer. *Neuroimage* **32**, 180–194 (2006).

37. Meier, L., Van De Geer, S. & Bühlmann, P. The group lasso for logistic regression. *J. R. Stat. Soc. B* **70**, 53–71 (2008).

38. Huang, K. et al. A multipredictor model to predict the conversion of mild cognitive impairment to Alzheimer's disease by using a predictive nomogram. *Neuropsychopharmacology* **45**, 358–366 (2020).

39. Khedher, L. et al. Independent component analysis-support vector machine-based computer-aided diagnosis system for Alzheimer's with visual support. *Int. J. Neural Syst.* **27**, 1650050 (2017).

40. Long, X., Chen, L., Jiang, C. & Zhang, L. Prediction and classification of Alzheimer disease based on quantification of MRI deformation. *PLoS ONE* **12**, e0173372 (2017).

41. Dimitriadis, S. I., Liparas, D. & Tsolaki, M. N. Random forest feature selection, fusion and ensemble strategy: combining multiple morphological MRI measures to discriminate among healhy elderly, MCI, cMCI and alzheimer's disease patients: from the alzheimer's disease neuroimaging initiative (ADNI) database. *J. Neurosci. Methods* **302**, 14–23 (2018).

42. Won, S. Y. et al. Quality reporting of radiomics analysis in mild cognitive impairment and Alzheimer's disease: a roadmap for moving forward. *Korean J. Radiol.* **21**, 1345–1354 (2020).

43. Popuri, K., Ma, D., Wang, L. & Beg, M. F. Using machine learning to quantify structural MRI neurodegeneration patterns of Alzheimer's disease into dementia score: independent validation on 8,834 images from ADNI, AIBL, OASIS, and MIRIAD databases. *Hum. Brain Mapp.* **41**, 4127–4147 (2020).

44. Bateman, R. J. et al. Clinical and biomarker changes in dominantly inherited Alzheimer's disease. *N. Engl. J. Med.* **367**, 795–804 (2012).

45. Liu, J., Wang, J., Hu, B., Wu, F. X. & Pan, Y. Alzheimer's disease classification based on individual hierarchical networks constructed with 3-D texture features. *IEEE Trans. Nanobioscience* **16**, 428–437 (2017).

46. de Vos, F. et al. Combining multiple anatomical MRI measures improves Alzheimer's disease classification. *Hum. Brain Mapp.* **37**, 1920–1929 (2016).

47. Bartos, A., Gregus, D., Ibrahim, I. & Tintěra, J. Brain volumes and their ratios in Alzheimer´s disease on magnetic resonance imaging segmented using Freesurfer 6.0. *Psychiatry Res. Neuroimaging* **287**, 70–74 (2019).

48. Arendt, T., Brückner, M. K., Morawski, M., Jäger, C. & Gertz, H.-J. Early neurone loss in Alzheimer's disease: cortical or subcortical? *Acta Neuropathol. Commun.* **3**, 10 (2015).

49. Thompson, P. M. et al. Cortical change in Alzheimer's disease detected with a disease-specific population-based brain atlas. *Cereb. Cortex* **11**, 1–16 (2001).

50. Fjell, A. M. et al. What is normal in normal aging? Effects of aging, amyloid and Alzheimer's disease on the cerebral cortex and the hippocampus. *Prog. Neurobiol.* **117**, 20–40 (2014).

51. Barbará-Morales, E., Pérez-González, J., Rojas-Saavedra, K. C. & Medina-Bañuelos, V. Evaluation of brain tortuosity measurement for the automatic multimodal classification of subjects with Alzheimer's disease. *Comput. Intell. Neurosci.* **2020**, 4041832–4041832 (2020).

52. Apostolova, L. G. et al. Hippocampal atrophy and ventricular enlargement in normal aging, mild cognitive impairment (MCI), and Alzheimer Disease. *Alzheimer Dis. Assoc. Disord.* **26**, 17–27 (2012).

53. Rudelli, R. D., Ambler, M. W. & Wisniewski, H. M. Morphology and distribution of Alzheimer neuritic (senile) and amyloid plaques in striatum and diencephalon. *Acta Neuropathol.* **64**, 273–281 (1984).

54. Toniolo, S. et al. Cerebellar white matter disruption in Alzheimer's disease patients: a diffusion tensor imaging study. *J. Alzheimer's Dis.* **74**, 615–624 (2020).

55. Farrer, L. A. et al. Effects of age, sex, and ethnicity on the association between apolipoprotein E genotype and Alzheimer disease. A meta-analysis. APOE and Alzheimer disease meta analysis consortium. *JAMA* **278**, 1349–1356 (1997).

56. Osherovich, L. TOMMorrow's AD marker. *Science-Business eXchange* **2**, 1165–1165 (2009).

57. Yu, C. E. et al. Comprehensive analysis of APOE and selected proximate markers for late-onset Alzheimer's disease: patterns of linkage disequilibrium and disease/marker association. *Genomics* **89**, 655–665 (2007).

58. Falahati, F. et al. The effect of age correction on multivariate classification in Alzheimer's disease, with a focus on the characteristics of incorrectly and correctly classified subjects. *Brain Topogr.* **29**, 296–307 (2016).

59. Moradi, E., Pepe, A., Gaser, C., Huttunen, H. & Tohka, J. Machine learning framework for early MRI-based Alzheimer's conversion prediction in MCI subjects. *Neuroimage* **104**, 398–412 (2015).

60. Ammari, S. et al. Influence of magnetic field strength on magnetic resonance imaging radiomics features in brain imaging, an in vitro and in vivo study. *Front. Oncol.* **10**, 541663 (2021).

61. Inglese, M. et al. Mesoscopic architecture of living Alzheimer's disease brain revealed, Mendeley Data, V1. (2022).

62. DeLong, E. R., DeLong, D. M. & Clarke-Pearson, D. L. Comparing the areas under two or more correlated receiver operating characteristic curves: a nonparametric approach. *Biometrics* **44**, 837–845 (1988).

## Acknowledgements
The radiomics analysis was funded by Imperial College London NIHR Biomedical Research Centre. E.O.A. acknowledges support from UK Medical Research Council Grant MR/N020782. MRI data collection and sharing for this project was funded by the Alzheimer's Disease Neuroimaging Initiative (ADNI) (National Institutes of Health Grant U01 AG024904) and DOD ADNI (Department of Defense award number W81XWH-12-2-0012). ADNI is funded by the National Institute on Aging, the National Institute of Biomedical Imaging and Bioengineering, and through generous contributions from the following: AbbVie, Alzheimer's Association; Alzheimer's Drug Discovery Foundation; Araclon Biotech; BioClinica, Inc.; Biogen; Bristol-Myers Squibb Company; CereSpir, Inc.; Cogstate; Eisai Inc.; Elan Pharmaceuticals, Inc.; Eli Lilly and Company; EuroImmun; F. Hoffmann-La Roche Ltd and its affiliated company Genentech, Inc.; Fujirebio; GE Healthcare; IXICO Ltd.; Janssen Alzheimer Immunotherapy Research & Development, LLC.; Johnson & Johnson Pharmaceutical Research & Development LLC.; Lumosity; Lundbeck; Merck & Co., Inc.; Meso Scale Diagnostics, LLC.; NeuroRx Research; Neurotrack Technologies; Novartis

Pharmaceuticals Corporation; Pfizer Inc.; Piramal Imaging; Servier; Takeda Pharmaceutical Company; and Transition Therapeutics. The Canadian Institutes of Health Research is providing funds to support ADNI clinical sites in Canada. Private sector contributions are facilitated by the Foundation for the National Institutes of Health (www.fnih.org). The grantee organisation is the Northern California Institute for Research and Education, and the study is coordinated by the Alzheimer's Therapeutic Research Institute at the University of Southern California. ADNI data are disseminated by the Laboratory for Neuro Imaging at the University of Southern California.

## Author contributions
E.O.A. conceived the project. M.I. designed the project, collected data and provided computational analysis. K.L.-R. provided bioinformatics analysis. R.J.P., N.P., F.L., Z.W. and C.C. collected clinical data. M.G.-S., W.R.C., H.L., P.A.M., N.P., E.O.A. and M.I. contributed to the interpretation of data. M.I. and E.O.A. wrote the manuscript. All authors edited the manuscript. **ADNI:** Data used in the preparation of this article were obtained from the Alzheimer's Disease Neuroimaging Initiative (ADNI) database (adni.loni.usc.edu). As such, the investigators within the ADNI contributed to the design and implementation of ADNI and/or provided data but did not participate in the analysis or writing of this report.

## Competing interests
The authors declare no competing interests.

## Additional information

## the Alzheimer's Disease Neuroimaging Initiative

Lisa C. Silbert[8], Betty Lind[8], Rachel Crissey[8], Jeffrey A. Kaye[8], Raina Carter[8], Sara Dolen[8], Joseph Quinn[8], Lon S. Schneider[9], Sonia Pawluczyk[9], Mauricio Becerra[9], Liberty Teodoro[9], Karen Dagerman[9], Bryan M. Spann[9], James Brewer[10], Helen Vanderswag[10], Adam Fleisher[10], Jaimie Ziolkowski[11], Judith L. Heidebrink[11], Zbizek-Nulph[11], Joanne L. Lord[11], Lisa Zbizek-Nulph[11], Ronald Petersen[12], Sara S. Mason[12], Colleen S. Albers[12], David Knopman[12], Kris Johnson[12], Javier Villanueva-Meyer[13], Valory Pavlik[13], Nathaniel Pacini[13], Ashley Lamb[13], Joseph S. Kass[13], Rachelle S. Doody[13], Victoria Shibley[13], Munir Chowdhury[13], Susan Rountree[13], Mimi Dang[13], Yaakov Stern[14], Lawrence S. Honig[14], Akiva Mintz[14], Beau Ances[15], John C. Morris[15], David Winkfield[15], Maria Carroll[15], Georgia Stobbs-Cucchi[15], Angela Oliver[15], Mary L. Creech[15], Mark A. Mintun[15], Stacy Schneider[15], David Geldmacher[16], Marissa Natelson Love[16], Randall Griffith[16], David Clark[16], John Brockington[16], Daniel Marson[16], Hillel Grossman[17], Martin A. Goldstein[17], Jonathan Greenberg[17], Effie Mitsis[17], Raj C. Shah[18], Melissa Lamar[18], Ajay Sood[18], Kimberly S. Blanchard[18], Debra Fleischman[18], Konstantinos Arfanakis[18], Patricia Samuels[18], Ranjan Duara[19], Maria T. Greig-Custo[19], Rosemarie Rodriguez[19], Marilyn Albert[20], Daniel Varon[20], Chiadi Onyike[20], Leonie Farrington[20], Scott Rudow[20], Rottislav Brichko[20], Maria T. Greig[20], Stephanie Kielb[20], Amanda Smith[21], Balebail Ashok Raj[21], Kristin Fargher[21], Martin Sadowski[22], Thomas Wisniewski[22], Melanie Shulman[22], Arline Faustin[22], Julia Rao[22], Karen M. Castro[22], Anaztasia Ulysse[22], Shannon Chen[22], Mohammed O. Sheikh[22], Jamika Singleton-Garvin[22], P. Murali Doraiswamy[23], Jeffrey R. Petrella[23], Olga James[23], Terence Z. Wong[23], Salvador Borges-Neto[23], Jason H. Karlawish[24], David A. Wolk[24], Sanjeev Vaishnavi[24], Christopher M. Clark[24], Steven E. Arnold[24], Charles D. Smith[25], Gregory A. Jicha[25], Riham El Khouli[25], Flavius D. Raslau[25], Oscar L. Lopez[26], Michelle Zmuda[26], Meryl Butters[26], MaryAnn Oakley[26], Donna M. Simpson[26], Anton P. Porsteinsson[27], Kim Martin[27], Nancy Kowalski[27], Kimberly S. Martin[27], Melanie Keltz[27], Bonnie S. Goldstein[27], Kelly M. Makino[27], M. Saleem Ismail[27], Connie Brand[27], Christopher Reist[28], Gaby Thai[28], Aimee Pierce[28], Beatriz Yanez[28], Elizabeth Sosa[28], Megan Witbracht[28], Brendan Kelley[29], Trung Nguyen[29], Kyle Womack[29], Dana Mathews[29], Mary Quiceno[29],

Allan I. Levey[30], James J. Lah[30], Ihab Hajjar[30], Janet S. Cellar[30], Jeffrey M. Burns[31], Russell H. Swerdlow[31], William M. Brooks[31], Daniel H. S. Silverman[32], Sarah Kremen[32], Liana Apostolova[32], Kathleen Tingus[32], Po H. Lu[32], George Bartzokis[32], Ellen Woo[32], Edmond Teng[32], Neill R. Graff-Radford[33], Francine Parfitt[33], Kim Poki-Walker[33], Martin R. Farlow[34], Ann Marie Hake[34], Brandy R. Matthews[34], Jared R. Brosch[34], Scott Herring[34], Christopher H. van Dyck[35], Adam P. Mecca[35], Susan P. Good[35], Martha G. MacAvoy[35], Richard E. Carson[35], Pradeep Varma[35], Howard Chertkow[36], Susan Vaitekunis[36], Chris Hosein[36], Sandra Black[37], Bojana Stefanovic[37], Chris Chinthaka Heyn[37], Ging-Yuek Robin Hsiung[38], Ellen Kim[38], Benita Mudge[38], Vesna Sossi[38], Howard Feldman[38], Michele Assaly[38], Elizabeth Finger[39], Stephen Pasternak[39], Irina Rachinsky[39], Andrew Kertesz[39], Dick Drost[39], John Rogers[39], Ian Grant[40], Brittanie Muse[40], Emily Rogalski[40], Jordan Robson M. -Marsel Mesulam[40], Diana Kerwin[40], Chuang-Kuo Wu[40], Nancy Johnson[40], Kristine Lipowski[40], Sandra Weintraub[40], Borna Bonakdarpour[40], Nunzio Pomara[41], Raymundo Hernando[41], Antero Sarrael[41], Howard J. Rosen[42], Scott Mackin[42], Craig Nelson[42], David Bickford[42], Yiu Ho Au[42], Kelly Scherer[42], Daniel Catalinotto[42], Samuel Stark[42], Elise Ong[42], Dariella Fernandez[42], Bruce L. Miller[42], Howard Rosen[42], David Perry[42], Raymond Scott Turner[43], Kathleen Johnson[43], Brigid Reynolds[43], Kelly MCCann[43], Jessica Poe[43], Reisa A. Sperling[44], Keith A. Johnson[44], Gad A. Marshall[44], Jerome Yesavage[45], Joy L. Taylor[45], Steven Chao[45], Jaila Coleman[45], Jessica D. White[45], Barton Lane[45], Allyson Rosen[45], Jared Tinklenberg[45], Christine M. Belden[46], Alireza Atri[46], Bryan M. Spann[46], Kelly A. Clark Edward Zamrini[46], Marwan Sabbagh[46], Ronald Killiany[47], Robert Stern[47], Jesse Mez[47], Neil Kowall[47], Andrew E. Budson[47], Thomas O. Obisesan[48], Oyonumo E. Ntekim[48], Saba Wolday[48], Javed I. Khan[48], Evaristus Nwulia[48], Sheeba Nadarajah[48], Alan Lerner[49], Paula Ogrocki[49], Curtis Tatsuoka[49], Parianne Fatica[49], Evan Fletcher[50], Pauline Maillard[50], John Olichney[50], Charles DeCarli[50], Owen Carmichael[50], Vernice Bates[51], Horacio Capote[51], Michelle Rainka[51], Michael Borrie[52], T. -Y Lee[52], Rob Bartha[52], Sterling Johnson[53], Sanjay Asthana[53], Cynthia M. Carlsson[53], Allison Perrin[54], Anna Burke[54], Douglas W. Scharre[55], Maria Kataki[55], Rawan Tarawneh[55], Brendan Kelley[55], David Hart[56], Earl A. Zimmerman[56], Dzintra Celmins[56], Delwyn D. Miller[57], Laura L. Boles Ponto[57], Karen Ekstam Smith[57], Hristina Koleva[57], Hyungsub Shim[57], Ki Won Nam[57], Susan K. Schultz[57], Jeff D. Williamson[58], Suzanne Craft[58], Jo Cleveland[58], Mia Yang[58], Kaycee M. Sink[58], Brian R. Ott[59], Jonathan Drake[59], Geoffrey Tremont[59], Lori A. Daiello[59], Jonathan D. Drake[59], Marwan Sabbagh[60], Aaron Ritter[60], Charles Bernick[60], Donna Munic[60], Akiva Mintz[60], Abigail O'Connelll[61], Jacobo Mintzer[61], Arthur Wiliams[61], Joseph Masdeu[62], Jiong Shi[63], Angelica Garcia[63], Marwan Sabbagh[63], Paul Newhouse[64], Steven Potkin[65], Stephen Salloway[66], Paul Malloy[66], Stephen Correia[66], Smita Kittur[67], Godfrey D. Pearlson[68], Karen Blank[68], Karen Anderson[68], Laura A. Flashman[69], Marc Seltzer[69], Mary L. Hynes[69], Robert B. Santulli[69], Norman Relkin[70], Gloria Chiang[70], Michael Lin[70], Lisa Ravdin[70], Athena Lee[70], Carl Sadowsky[71], Walter Martinez[71], Teresa Villena[71], Elaine R. Peskind[72], Eric C. Petrie[72] & Gail Li[72]

[8]Oregon Health & Science University, Portland, OR, USA. [9]University of Southern California, Los Angeles, CA, USA. [10]University of California, San Diego, CA, USA. [11]University of Michigan, Ann Arbor, MI, USA. [12]Mayo Clinic, Rochester, NY, USA. [13]Baylor College of Medicine, Houston, TX, USA. [14]Columbia University Medical Center, New York, NY, USA. [15]Washington University, St. Louis, MO, USA. [16]University of Alabama at Birmingham, Birmingham, AL, USA. [17]Mount Sinai School of Medicine, New York, NY, USA. [18]Rush University Medical Center, Chicago, IL, USA. [19]Wien Center, Wien, Austria. [20]Johns Hopkins University, Baltimore, MD, USA. [21]University of South Florida, USF Health Byrd Alzheimer's Institute, Tampa, FL, USA. [22]New York University, New York, NY, USA. [23]Duke University Medical Center, Durham, NC, USA. [24]University of Pennsylvania, Philadelphia, PA, USA. [25]University of Kentucky, Lexington, KY, USA. [26]University of Pittsburgh, Pittsburgh, PA, USA. [27]University of Rochester Medical Center, Rochester, NY, USA. [28]University of California Irvine IMIND, Irvine, CA, USA. [29]University of Texas Southwestern Medical School, Dallas, TX, USA. [30]Emory University, Atlanta, GA, USA. [31]University of Kansas, Medical Center, Kansas City, KS, USA. [32]University of California, Los Angeles, CA, USA. [33]Mayo Clinic, Jacksonville, FL, USA. [34]Indiana University, Bloomington, IL, USA. [35]Yale University School of Medicine, New Haven, CT, USA. [36]McGill University, Montreal-Jewish General Hospital, Montreal, QC, Canada. [37]Sunnybrook Health Sciences, Ontario, ON, Canada. [38]U.B.C. Clinic for AD & Related Disorders, Vancouver, BC, Canada. [39]St. Joseph's Health Care, Hamilton, ON, Canada. [40]Northwestern University, Evanston, IL, USA. [41]Nathan Kline Institute, New York, NY, USA. [42]University of California, San Francisco, CA, USA. [43]Georgetown University Medical Center,

Georgetown, DC, USA. [44]Brigham and Women's Hospital, Boston, MA, USA. [45]Stanford University, Stanford, CA, USA. [46]Banner Sun Health Research Institute, Sun City, AZ, USA. [47]Boston University, Boston, MA, USA. [48]Howard University, Washington, WA, USA. [49]Case Western Reserve University, Cleveland, OH, USA. [50]University of California, Sacramento, CA, USA. [51]Dent Neurologic Institute, Amherst, MA, USA. [52]Parkwood Institute, London, ON, Canada. [53]University of Wisconsin, Madison, WI, USA. [54]Banner Alzheimer's Institute, Phoenix, AZ, USA. [55]Ohio State University, Columbus, OH, USA. [56]Albany Medical College, Albany, NY, USA. [57]University of Iowa College of Medicine, Iowa City, IA, USA. [58]Wake Forest University Health Sciences, Winston-Salem, NC, USA. [59]Rhode Island Hospital, Providence, RI, USA. [60]Cleveland Clinic Lou Ruvo Center for Brain Health, Las Vegas, NV, USA. [61]Roper St. Francis Healthcare, Charleston, SC, USA. [62]Houston Methodist Neurological Institute, Houston, TX, USA. [63]Barrow Neurological Institute, Phoenix, AZ, USA. [64]Vanderbilt University Medical Center, Nashville, TN, USA. [65]Long Beach VA, Long Beach, CA, USA. [66]Butler Hospital Memory and Aging Program, Butler Hospital, Providence, RI, USA. [67]Neurological Care of CNY, Syracuse, NY, USA. [68]Hartford Hospital, Olin Neuropsychiatry Research Center, Hartford, CT, USA. [69]Dartmouth-Hitchcock Medical Center, Lebanon, PA, USA. [70]Cornell University, Ithaca, NY, USA. [71]Premiere Research Institute, Palm Beach Neurology, Palm Beach, FL, USA. [72]University of Washington, Washington, WA, USA.

