## [Peer Review File · Communications Medicine]

Reviewers' comments:

Reviewer #1 (Remarks to the Author):

The ms with title 'Discernment of the mesoscopic architecture of living Alzheimer's disease brain aids disease detection' proposed a predictive model that computes multi-regional statistical morpho-functional mesoscopic traits from T1-weighted MRI scans, with or without cognitive scores. For each patient, a biomarker called "Alzheimer's Predictive Vector" (ApV) was derived. The ApV reliably discriminates between subjects with (ADrp) and without (nADrp) Alzheimer's related pathologies (98% and 81% accuracy between ADrp. This new test is found to be superior to hippocampal atrophy and CSF A-beta measure. The report is well written, but the reviewer has some concerns as given below.

Introduction:

There are some statements which need attention.

- 1) 'The disease is characterized by three main events...'. This is about A/T/N framework. Two things the authors should provide more clarifications. First, for 'N', atrophy of grey matter is only one among a number of others. For example, the hypometabolism measured by FDG-PET N. Second, the author may note a recent publication on the addition of inflammation (nature medicine: Microglial activation and tau propagate jointly across Braak stages).
- 2) The CSF biomarker was mentioned and also used (ApVs) but not blood based biomarkers (BBB) which have been a very hot topic in recent years especially with much increased sensitivity and accuracy. This BBB omission should be dealt with, and the imaging (especially T1-MRI) advantages should be discussed in the context of the current advances of BBB. This is not to understate the importance of widely used T1-MRI (as a standard of care) and its novel use as proposed in this study, but the new T1-MRI based approach, its feasibility/value should be discussed.
- 3) In addition to CSF which was mentioned, PET is probably the best for measuring amyloid and tau (while BBB is very promising down the road). The 'real-world' data actually had this included to define AMY+/-.
- 4) The ADrp (AD related pathology) was defined in the Methods section. This is different from commonly used amyloid and also tau characterization (under the ATN framework). It is probably better to clarify this difference very early on. This is especially concerning, as shown in Supplementary Figure 4 for IMC cohort, that there is no difference between AMY- and AMY+ by ApV1

Results

- 1) It is new info to this reviewer with regard to the number of PD and FTD (94 patients with FTD, and 84 with PD) from ADNI. It is good to know but also wondering if the authors can check if they are comorbidity (with AD) or pure PD/FDT?
- 2) Are the subjects from OASIS all normal? Their diagnosis information should be provided together with number of subjects in each diagnostic group (or in light of ADrp).
- 3) The same for ADNI3 data (with 3T)
- 4) Can the authors define the 'early form' of AD from the 'real-world' cohort (IMC)? Is it the population with disease (AD) causing genes such as APP, PS-1 and PS-2 or what? Look like it is not the case (so not early onset of AD)
- 5) The performance on the unseen ADNI 3T data was poor. This was acknowledged in the Discussion. However, this might be something to test out quickly as 3T data are widely available, from ADNI for example
- 6) For GWAS results, how about APOE4?

Methods:

1) The use of FreeSurfer will make the preprocessing process very long (or with HPC). Can authors comment on this?

The reviewer acknowledges that some of the comments/concerns were well addressed in the Discussion section of the manuscript (AD pathology, e.g., additional discussion of the 'real-world' dataset and its nature, etc.)

Reviewer #2 (Remarks to the Author):

Discernment of the mesoscopic architecture of living Alzheimer's disease brain aids disease detection

Summary

In this work, the authors develop a model to predict Alzheimer's disease (AD) pathology from "morpho-functional mesoscopic traits", which turn out to be T1w morphological characteristics, and specifically radiomics features from FreeSurfer parcellations. On those features a model is trained to: i) predict AD related pathology and ii) distinguish MCI-AD from AD patients. A "LASSO" approach is performed to evaluate features' importance and select only the one contributing to the model. Moreover, an Alzheimer's Predictive Vector (ApV) was computed from the weighted sum of the selected features.

14 regions were selected from the first trained model discriminating AD pathology. The ApV derived from these regions (also when accounting for cognitive scores) showed higher specificity, sensitivity and accuracy than other traditional biomarkers. The second model selected 8 features of interest. Also in this case, the derived ApV showed higher accuracy and sensitivity compared to other biomarkers. The models were further tested on real world data. Eventually, the authors relate the ApV features with GWAS analysis and DTI data. Claims that this work would make neuroradiology reads unnecessary and beating performance of other clinically used test in isolation are premature and unfounded.

Strengths of the paper

The results reported in the paper are certainly of interest for the field, and show a possible application of automated algorithms to "real life" AD diagnostic problems. The method presented in the manuscript are rigorous and represent an interesting combination of statistical methodologies (e.g. features selection and univariate prediction) to obtain a better prediction of the pathology. The authors fail to explain what is different from support vector regression or similar techniques.

Limitations

Overall, the paper looks chaotic in terms of structure and it is sometimes hard for the reader to follow and grasp the message being conveyed. I think this work would benefit from a general restructuring of some paragraphs to make the aim clearer. Interpretation and critical thinking of the results might also be improved in the discussion section. Few methodological issues need also to be addressed.

More specific comments about these and other points can be found in the next sections.

Major Comments

- Abstract: I believe the abstract does not give a clear overview of the hypothesis and methodological procedures of the work. For example: there is no description of the sample used; The ApV is mentioned but no explanation of how this is derived is given (LASSO and radiomics are not mentioned); fractional anisotropy is only introduced in the final sentence and it is not clear explanation of the analysis this is used for.

- Introduction: The introduction section is rather weak in giving the reader an overview of the specific concepts needed for this paper. The first two paragraphs give a quite general overview of AD prevalence and characteristics which I believe could be shortened and focus on the only concepts that are really related to this work (e.g. structural MRI, radiomics). The main scope of the paper becomes clear only in the third paragraph "Critically, feature extracted from ...". I would advise a general restructuring of the introduction to convey a much clearer and strong message to the reader.

- Methods, AvP: The creation of the univariate vector from the set of selected features is surely an interesting step taken by the authors. I'm wondering whether using multivariate prediction procedures on the reduced (or full) set of extracted features would be beneficial compared to the currently used univariate technique. It would be interesting to see the comparison between the two methods.

- Methods, LASSO: Was there any hyperparameter optimization for LASSO feature selection? Please clarify.

- Methods, FreeSurfer: Why were only white matter and subcortical regions used? Why not cortical parcellation?

- Fractional anisotropy Analysis: Overall, the analysis of the fractional anisotropy is interesting but not connected to the rest of the paper. While reading the manuscript this is unexpected and there is no clear explanation of how this relate to the general scope of the paper. How does this relate with the trained model and automatic prediction of pathology? As this is an interesting analysis, I would advise the author to make this clearer already from the introduction (following my previous comments on it)

- Results, Patients characteristics: At the moment this paragraph appears chaotic and does not allow for a clear understanding of the final sample used. Some of the groups are introduced in the text (like the training group, but also the 'real world') while others are introduced in a numeric list (1. An unseen ..). I would advise the authors to follow the same structure as in the method section, even by referring to it, which is more straightforward.

- The 'real-world': There is no specification throughout the manuscript what makes this dataset a 'real-world' dataset. The results paragraph about the model performance on this dataset is incomplete, only stating that the model "classified subjects as being nADrp and MCI-AD" or subjects were "mainly classifies as ..". This section is therefore missing statistics.

- Discussion: I believe this section would benefit of a deeper discussion of the selected features (which are never mentioned in the paper) and their possible meaning for AD, and a critical

evaluation of the applicability of this method in real medical contexts.

Minor Comments

- Please check the font used throughout the text as it is sometime changing
- Please check that abbreviations are always introduced at their first appearance in the manuscript
- Line 36: "for this reason" is actually connecting two clauses that are not causally linked (how does early amyloid relate to structural changes?)
- Line 173: It seems that there is a "with" in the wrong position, probably to be deleted
- Line 187: "applied unmodified to a different field strength", it would probably better to be more specific and bring the reader attention back by specifying the 3T scanner.
- Line 226: "As before" seems unnecessary
- Line 312: "confirming results ..." seems to be better suited for a discussion section

hope these comments are useful

Frederik Barkhof

Reviewer #3 (Remarks to the Author):

Review: Nature Communications in Medicine (New Journal)

Title: Discernment of the mesoscopic architecture of living Alzheimer's disease brain aids disease detection

Summary: The authors developed a novel predictive model to classify patients with late-onset Alzheimer's disease. They use their novel unsupervised method to derive an "Alzheimer's Predictive Vector" that can distinguish between patients with Alzheimer related pathologies and without Alzheimer related pathologies. The authors claim that their method based on MRI imaging outperforms fluid biomarker measures and atrophy measures of the hippocampus.

A single-variant association study was performed to identify markers associated with the novel disease classifier. Markers in TOMM40 at the APOE locus were significantly associated with patients that were determined to show Alzheimer's related pathologies.

While this manuscript describes an interesting approach to classify patients based on MRI scans and infers genetic associations using this novel phenotype, there is an overall lack of clarity concerning the experimental design and interpretation of the results. My major criticisms are listed below:

1.) The authors performed single-variant association analysis based on multiple novel phenotype which was inferred using their LASSO approach. They report statistical significance based on a WALD test, which is commonly used to extract p-values from a logistic regression model. It remains unclear what was compared in their genome-wide association study and how the analysis was performed. The authors also need to state what type of quality controls were performed on the data. The results report a genome-wide significant variant in TOMM40, which is at the APOE locus, the most significant risk factor for AD. This is to be expected when comparing AD cases with cognitively normal elderly controls. The authors need to clarify their statistical model for their GWAs analysis, provide a detailed overview of the variant association at the locus and adjust for APOE status when performing their single-variant association. Adjusting for APOE status, will allow the authors to determine whether this is a true independent association with variants in TOMM40 or an association signal purely driven by the APOE4 allele and surrounding variants in LD.

2.) Several previous studies have used ADNI imaging data to classify patients into distinct subgroups and predict conversion from MCI to AD. These studies reported high accuracy and similar prediction rates as reported in this manuscript. While several studies have used lasso or elastic net, others have used transfer learning methods for this task. These previous studies have provided proof of principle that this approach can be used to perform genetic mapping to identify markers associated with different imaging related phenotypes. The manuscript lacks a more detailed comparison to other methods, highlighting the novelty of their approach. The authors need to state in which cases their method outperforms existing methods in classifying AD patients.

3.) The authors continuously overstate their findings. Language such as “state-of-the-art accuracy” (lines 332.f) or “high accuracy is remarkable” (lines 238.f) should be avoided.

As noted in the previous point 2, the reported accuracy is comparable to previous studies (PMID: 27054198). In addition, the authors state that they used external data sets to confirm the results and that their method is robust. However, it does not become clear how for example patients from the OASIS cohort were selected for their analysis and compared to the results from the ADNI data set.

4.) The current structure and organization of the manuscript makes it difficult to read. Figures and figure captions are placed throughout the manuscript, and it becomes difficult for the reader to distinguish between the text in the manuscript and the figure captions. Some of the plots included in the figures, such as the Q-Q plot or the Manhattan plot don't add additional value to the results section and should be moved to the supplements. Instead, a locus zoom plot highlighting the variant association at the APOE4 locus would be more helpful for the reader to interpret the results from the single variant association. Furthermore, a lot of important information is hidden in the supplementary notes (1+2) and at least the discussion related to the genetic association should be move to the main text. Also, supplement tables 1 and 2 either not formatted correctly or the pdf conversion from text did not work as intended.

Dear Editor,

we would like to thank the reviewers for the detailed comments and further suggestions for the manuscript. After the completion of the suggested new edits, we believe that the revised manuscript has benefitted from an improvement in overall presentation and clarity.

Below, you will find a point-by-point description of how each comment was addressed in the manuscript. Original reviewer comments in regular typeface, responses in italic.

Reviewer #1 (Remarks to the Author):

The ms with title ‘Discernment of the mesoscopic architecture of living Alzheimer’s disease brain aids disease detection’ proposed a predictive model that computes multi-regional statistical morpho-functional mesoscopic traits from T1-weighted MRI scans, with or without cognitive scores. For each patient, a biomarker called “Alzheimer’s Predictive Vector” (ApV) was derived. The ApV reliably discriminates between subjects with (ADrp) and without (nADrp) Alzheimer’s related pathologies (98% and 81% accuracy between ADrp). This new test is found to be superior to hippocampal atrophy and CSF A-beta measure. The report is well written, but the reviewer has some concerns as given below.

Introduction:

There are some statements which need attention.

1) ‘The disease is characterized by three main events...’. This is about A/T/N framework. Two things the authors should provide more clarifications. First, for ‘N’, atrophy of grey matter is only one among a number of others. For example, the hypometabolism measured by FDG-PET N. Second, the author may note a recent publication on the addition of inflammation (nature medicine: Microglial activation and tau propagate jointly across Braak stages).

Thank you for your comment. The introduction has been modified as follows:

*The disease is characterized by several events. The National Institute on Aging and Alzheimer's Association has proposed a classification system to categorize individuals based on biomarker evidence of pathology. This is called ATN classification system and is used to rate subjects for the presence of β -amyloid (CSF A β or amyloid PET: “A”), hyperphosphorylated tau (CSF p-tau or tau PET: “T”), and neurodegeneration (atrophy on structural MRI, FDG PET, or CSF total tau: “N”), resulting in 8 possible biomarker combinations [Ebenau JL, Timmers T, Wesselman LMP, et al. ATN classification and clinical progression in subjective cognitive decline: The SCIENCE project. *Neurology*. 2020;95(1):e46-e58.*

*doi:10.1212/WNL.0000000000009724]. Furthermore, recent report of the involvement of microglial activation in the spread of tau tangles over the neocortex in AD suggests an additional inflammation biomarker for AD [Pascoal, T.A., Benedet, A.L., Ashton, N.J. et al. Microglial activation and tau propagate jointly across Braak stages. *Nat Med* 27, 1592–1599 (2021). <https://doi.org/10.1038/s41591-021-01456-w>].*

Lines 45-52.

2) The CSF biomarker was mentioned and also used (ApVs) but not blood based biomarkers (BBB) which have been a very hot topic in recent years especially with much increased sensitivity and accuracy. This BBB omission should be dealt with, and the imaging (especially T1-MRI) advantages should be discussed in the context of the current advances of BBB. This is not to understate the importance of widely used T1-MRI (as a standard of care) and its novel use as proposed in this study, but the new T1-MRI based approach, its feasibility/value should be discussed.

Thank you for this comment. We commented on BBB biomarkers in the Introduction section adding the strength and limitations of this approach. We wrote:

(...) Similarly, blood based biomarkers (BBB), which are eagerly awaited to address issues related to the invasiveness and high cost of CSF-based ones, often stall in the early stages because of a disconnect between academia, where biomarkers are identified, and industry, where they should be developed and commercially distributed [Hampel, H., O'Bryant, S.E., Molinuevo, J.L. et al. Blood-based biomarkers for Alzheimer disease: mapping the road to the clinic. *Nat Rev Neurol* **14**, 639–652 (2018). <https://doi.org/10.1038/s41582-018-0079-7>].

Lines 61-64.

3) In addition to CSF which was mentioned, PET is probably the best for measuring amyloid and tau (while BBB is very promising down the road). The 'real-world' data actually had this included to define AMY+/-.

Agreed. This cohort of patients underwent clinical amyloid PET imaging at the Imperial Memory Centre (IMC, London, UK) as part of their diagnostic workup (IMC cohort).

4) The ADrp (AD related pathology) was defined in the Methods section. This is different from commonly used amyloid and also tau characterization (under the ATN framework). It is probably better to clarify this difference very early on. This is especially concerning, as shown in Supplementary Figure 4 for IMC cohort, that there is no difference between AMY- and AMY+ by ApV1.

Thank you for your comment. This has been clarified in the "Characteristics of data and patients" paragraph as follows:

(...)They were grouped as 216 healthy controls, 208 subjects with MCI due to AD (referred to as MCI_{AD} in the text), 181 AD, 94 patients with Frontotemporal Dementia (FTD), and 84 with Parkinson's disease (PD). In particular, based on the data obtained from the ADNI database, two new groups of subjects were defined: the nADrp group, which contains subjects who do not show any pathology related to AD (healthy controls, PD and FTD were included here) and the ADrp group which, on the contrary, contains subjects with MCI due to AD and AD patients. The method was externally tested on: (...)

Results

1) It is new info to this reviewer with regard to the number of PD and FTD (94 patients with FTD, and 84 with PD) from ADNI. It is good to know but also wondering if the authors can check if they are comorbidity (with AD) or pure PD/FDT?

This has been checked and they are pure PD/FTD.

2) Are the subjects from OASIS all normal? Their diagnosis information should be provided together with number of subjects in each diagnostic group (or in light of ADrp).

Thank you for your comment. This information has been added in the "Characteristics of data and patients" subsection (Results section), as well as in the Methods section.

In the Results section this has been revised as follows:

The method was externally tested on:

- 1) *An unseen 1.5T dataset obtained from the Open Access Series of Imaging Studied (OASIS) consortium (<https://www.oasis-brains.org/>) of 64 subjects for whom baseline T1w sequence, age and MMSE score were available (53 CN and 11 AD).*

- 2) *An unseen 3T dataset of 402 subjects obtained from the ADNI3 cohort for whom baseline T1w sequence, age, cognitive scores and CSF related biomarkers were available (172 CN, 161 MCI_{AD} and 69 AD).*

In the Methods section this has been revised as follows:

The method was tested on four cohorts:

- (1) *The unseen 1.5T ADNI cohort (30% of the entire 1.5T cohort, made of 65 CN, 62 MCI_{AD}, 54 AD, 28 FTD and 25 PD);*
- (2) *The unseen 1.5T dataset: 64 subjects obtained from the OASIS consortium with T1w MRI scan and MMSE score (53 CN and 11 AD);*
- (3) *The unseen 3T dataset: 402 subjects obtained from ADNI with T1w MRI scan, MMSE, LDELTOTAL, A β , tau and ptau (172 CN, 161 MCI_{AD} and 69 AD);*
- (4) *The 'real-world' memory clinic cohort (IMC cohort): 83 patients with Atypical Presentations who underwent clinical amyloid PET imaging as part of their diagnostic workup with a 1.5T T1w MRI scan (45 AMY- and 38 AMY+) and LDELTOTAL and MMSE scores (for a subgroup of 22 subjects: 11 AMY- and 11 AMY+).*

Lines 405-415

- 3) The same for ADNI3 data (with 3T)

This is included in the previous response.

- 4) Can the authors define the 'early form' of AD from the 'real-world' cohort (IMC)? Is it the population with disease (AD) causing genes such as APP, PS-1 and PS-2 or what? Look like it is not the case (so not early onset of AD)

Thank you for your comment. The IMC cohort generally consisted of people with earlier onset than typical AD, given that they were being scanned according to the appropriate use criteria proposed by Johnson and colleagues (Journal of Nuclear Medicine, 2013). They did not have autosomal dominant AD caused by mutations in APP, PS1 or PS2 although a number were tested for these (i.e., those with family histories and onset before the age of 60). Only two patients (out of the total number of amyloid positive subjects) underwent dementia panel genetic testing and no pathogenic mutations were detected.

- 5) The performance on the unseen ADNI 3T data was poor. This was acknowledged in the Discussion.

However, this might be something to test out quickly as 3T data are widely available, from ADNI for example.

The performance of the model, trained on 1.5T T1w MRI data (ADNI1 cohort), was tested on a subgroup of the ADNI3 cohort, which has been scanned at 3T. Appreciating the importance of testing our model on images acquired at a different field strength, we decided to expand this analysis on all the available baseline 3T T1w MRI scans of the ADNI3 cohort (172 CN, 161 MCI_{AD} and 69 AD). Results have been updated in the manuscript in Table 1A and B:

A

	Training 1.5T ADNI dataset		Unseen 1.5T ADNI dataset		Unseen 1.5T OASIS dataset		Unseen 3T ADNI dataset		Volume of hippocampus	A β
	ApV ₁	ApV _{1s}	ApV ₁	ApV _{1s}	ApV ₁	ApV _{1s}	ApV ₁	ApV _{1s}		
AUC	0.9981	0.9971	0.9786	0.9490	0.6706	0.6801	0.6533	0.5192	0.7790	0.5045
Threshold	0.0938	-0.1969	0.0938	-0.1969	0.0938	-0.1969	0.0938	-0.1969	-0.1132	192
Specificity	0.9818	0.9669	0.9831	0.9671	0.8868	0.9057	0.9127	0.8081	0.2273	0.0091
Sensitivity	0.9819	0.9780	0.9741	0.9310	0.4545	0.4545	0.1739	0.2304	0.2941	1
Accuracy	0.9836	0.9728	0.9786	0.9554	0.8125	0.8281	0.4900	0.4776	0.2626	0.6236
NPV	0.9855	0.9750	0.9748	0.9484	0.8868	0.8889	0.4524	0.4398	0.2227	1
PPV	0.9818	0.9709	0.9826	0.9558	0.4545	0.500	0.7272	0.6162	0.2996	0.6223
LR+	54.4364	29.5952	57.4741	28.3034	4.0151	4.8182	1.9942	1.2010	0.3806	1.0009
LR-	0.0149	0.0868	0.0263	0.0713	0.6151	0.6023	0.9050	0.9522	3.1059	0
Yi	0.9672	0.9450	0.9572	0.8981	0.3413	0.3602	0.0867	0.0385	-0.4786	0.0092
DOR	3653.4	1301.6	2184.6	396.9	6.5278	8	2.2035	1.2612	0.1225	NA

B

	Training 1.5T ADNI dataset		Unseen 1.5T ADNI dataset		Unseen 3T ADNI dataset		Volume of hippocampus	A β
	ApV ₂	ApV _{2s}	ApV ₂	ApV _{2s}	ApV ₂	ApV _{2s}		
AUC	0.8580	0.9656	0.7258	0.8983	0.5072	0.7111	0.5345	0.5
Threshold	0.3017	0.8184	0.3017	0.8184	0.3017	0.8184	-0.7827	192
Specificity	0.9863	0.9384	0.9516	0.9384	1	0.9875	0.3387	0
Sensitivity	0.5590	0.8583	0.5000	0.8583	0.0289	0.4347	0.7593	1
Accuracy	0.7875	0.9011	0.7863	0.8633	0.6296	0.8217	0.5345	0.4887
NPV	0.7200	0.8839	0.6860	0.8839	0.7061	0.8030	0.6176	NA
PPV	0.9726	0.9237	0.9000	0.9237	1	0.9375	0.5000	0.4887
LR+	40.8110	13.9230	10.3333	13.9230	NA	35.0000	1.1481	1

LR-	0.4471	0.1510	0.5254	0.1510	0.9710	0.5723	0.7108	NA
Yi	0.5454	0.7966	0.4516	0.7966	0.0289	0.4223	0.0980	0
DOR	91.2857	92.1790	19.6667	92.1790	NA	61.1538	1.6154	NA

6) For GWAS results, how about APOE4?

Thank you for this comment. This has been further investigated and results have been included in the “Results” section:

(...) Similarly, for all cognitively normal vs mild cognitive impairment no SNPs were above the threshold. Additionally, in the ApV group, ADrp vs AD, the $p < 10^{-6}$ SNP rs575606 was above a threshold of $p < 10^{-6}$ (Supplementary Figure 1). When performing a GWAS adjusting for the presence of one or two APOE4 alleles, no SNPs were identified as significantly associated with AD in any of the outcomes (Supplementary Figure 7).

Lines 254-258.

Methods:

1) The use of FreeSurfer will make the preprocessing process very long (or with HPC). Can authors comment on this?

Thank you for this comment. We addressed this issue as one of the limitations to this study. We wrote:

A further limitation to this study is related to the computational effort needed to pre-process the structural MRI data. The segmentation step performed by FreeSurfer’s recon-all function usually requires about 10/12 hours per subject. In this regard, to reduce computation time, we decided to re-run the analyses in parallel using 12 logical cores; a group of 10/15 scans were segmented with this latter approach in the same amount of time. Another possible future solution to minimize segmentation time in clinical practice, could be the extraction of a custom T1w-MRI-based template built from the chosen dataset (e.g., using the SPM DARTEL pipeline).

Lines 382 – 392

Reviewer #2 (Remarks to the Author):

Discernment of the mesoscopic architecture of living Alzheimer’s disease brain aids disease detection

Summary

In this work, the authors develop a model to predict Alzheimer’s disease (AD) pathology from “morpho-functional mesoscopic traits”, which turn out to be T1w morphological characteristics, and specifically radiomics features from FreeSurfer parcellations. On those features a model is trained to: i) predict AD related pathology and ii) distinguish MCI-AD from AD patients. A “LASSO” approach is performed to evaluate features’ importance and select only the one contributing to the model. Moreover, an Alzheimer’s Predictive Vector (ApV) was computed from the weighted sum of the selected features.

14 regions were selected from the first trained model discriminating AD pathology. The ApV derived from these regions (also when accounting for cognitive scores) showed higher specificity, sensitivity and accuracy than other traditional biomarkers. The second model selected 8 features of interest. Also in this case, the derived ApV showed higher accuracy and sensitivity compared to other biomarkers. The models were further tested on real world data. Eventually, the authors relate the ApV features with GWAS analysis and DTI data.

Claims that this work would make neuroradiology reads unnecessary and beating performance of other clinically used test in isolation are premature and unfounded.

Strengths of the paper

The results reported in the paper are certainly of interest for the field, and show a possible application of automated algorithms to “real life” AD diagnostic problems. The method presented in the manuscript are rigorous and represent an interesting combination of statistical methodologies (e.g. features selection and univariate prediction) to obtain a better prediction of the pathology. The authors fail to explain what is different from support vector regression or similar techniques.

Limitations

Overall, the paper looks chaotic in term of structure and it is sometime hard for the reader to follow and grasp the message being conveyed. I think this work would benefit from a general restructuring of some paragraphs to make the aim clearer. Interpretation and critical thinking of the results might also be improved in the discussion section. Few methodological issue need also to be addressed. More specific comments about these and other points can be found in the next sections.

Major Comments

- Abstract: I believe the abstract does not give a clear overview of the hypothesis and methodological procedures of the work. For example: there is no description of the sample used; The AvP is mentioned but no explanation of how this is derived is given (LASSO and radiomics are not mentioned); fractional anisotropy is only introduced in the final sentence and it is not clear explanation of the analysis this is used for.

Thank you for your comment.

In the Abstract we decided to use a different but equivalent nomenclature to refer to radiomics: the “multi-regional statistical morpho-functional mesoscopic traits from T1-weighted MRI scans” are the features we extracted from MRI data. The LASSO has been now mentioned in the Abstract together with the use of fractional anisotropy in the multiparametric analysis. Lines 28-41.

- Introduction: The introduction section is rather weak in giving the reader an overview of the specific concepts needed for this paper. The first two paragraphs give a quite general overview of AD prevalence and characteristics which I believe could be shortened and focus on the only concepts that are really related to this work (e.g. structural MRI, radiomics). The main scope of the paper becomes clear only in the third paragraph “Critically, feature extracted from ...”.

I would advise a general restructuring of the introduction to convey a much clearer and strong message to the reader.

Thank you for your comment. The Introduction has been now restructured and the first two paragraphs have been restructured. Lines 44-100.

- Methods, AvP: The creation of the univariate vector from the set of selected features is surely an interesting step taken by the authors. I’m wondering whether using multivariate prediction procedures on the reduced (or full) set of extracted features would be beneficial compared to the currently used univariate technique. It would be interesting to see the comparison between the two methods.

Thank you for your comment. The accuracy of the multivariate analysis on the full set of features extracted by our software is reported in the following table, in comparison to our univariate method. We applied the most common machine learning algorithms (Support Vector Machine, Random Forest, Logistic Regression, Naïve Bayes and K-Nearest Neighbors) on the 1.5 T dataset.

	nADrp vs ADrp		MCI _{AD} vs AD	
	train	test	train	test
Ensemble Bagged Trees	87.5	31.1	66.9	11.2
Naïve Bayes	76.9	62.0	62.1	11.0
KNN	85.0	74.1	60.3	11.1
SVM	87.9	30.9	66.5	10.0
Our method	98.3	97.8	78.7	78.6

Following also a comment given by Reviewer 2, these results have been included in the Discussion section (and in Supplementary Table 4). Lines 286-190.

- Methods, LASSO: Was there any hyperparameter optimization for LASSO feature selection? Please clarify.

Thank you for this comment. The least absolute shrinkage and selection operator was implemented using the `lassoglm` function in Matlab. Ten-fold cross-validation was performed to select lambda which yielded the minimum cross-validated mean squared error. `Lassoglm`, by default, estimates the largest value of Lambda that gives a non-null model. The only hyperparameter we optimised was the `DFmax`, which defines the maximum number of non-zero coefficients in the model. This was set to infinite (default), 20 and 10.

- Methods, FreeSurfer: Why were only white matter and subcortical regions used? Why not cortical parcellation?

The default analysis pipeline in FreeSurfer computes several parcellations of the cortical surface in anatomical regions. We used the `aparc+aseg` file which includes cortical parcels and subcortical regions. The list of the 115 brain regions extracted with FreeSurfer is summarized in Supplementary Table 7.

- Fractional anisotropy Analysis: Overall, the analysis of the fractional anisotropy is interesting but not connected to the rest of the paper. While reading the manuscript this is unexpected and there is no clear explanation of how this relate to the general scope of the paper. How does this relate with the trained model and automatic prediction of pathology? As this is an interesting analysis, I would advise the author to make this clearer already from the introduction (following my previous comments on it)

Thank you for this comment. As widely published in the literature, a significant variation of fractional anisotropy is present in the brain of subjects with AD. For this reason, we here decided to investigate FA variations in the regions extracted by our algorithm (which do not necessarily overlap with the regions usually associated to AD) and also to use FA to ease the interpretation of the extracted features (Supplementary material).

- Results, Patients characteristics: At the moment this paragraph appears chaotic and does not allow for a clear understanding of the final sample used. Some of the groups are introduced in the text (like the training group, but also the ‘real world’) while others are introduced in a numeric list (1. An unseen ..). I would advise the authors to follow the same structure as in the method section, even by referring to it, which is more straightforward.

Thank you for this comment. This section has been restructured and cohorts have been introduced using a numeric list. Lines 113-153.

- The ‘real-world’: There is no specification throughout the manuscript what makes this dataset a ‘real-world’ dataset. The results paragraph about the model performance on this dataset is incomplete, only stating that the model “classified subjects as being nADrp and MCI-AD” or subjects were “mainly classifies as ..”. This section is therefore missing statistics.

Thank you for this comment. This cohort of subjects has been called ‘real-world’ group because it contains subjects who underwent clinical amyloid PET scan at our institution (“real world” to highlight the difference between them and the “clean” datasets downloaded from online available databases). The details about this cohort are summarised in the flowchart in Supplementary Figure 1. Furthermore in response to comments from Reviewer 1, we have included the following: The IMC cohort generally consisted of people with earlier onset than typical AD, given that they were being scanned according to the appropriate use criteria proposed by Johnson and colleagues (Journal of Nuclear Medicine, 2013).

The results paragraph about the model performance on this dataset refers to the results shown in Supplementary Figure 4. The paragraph has been modified as follows:

The model was tested on the IMC cohort, which includes subjects who underwent clinical amyloid PET scan at our institution and are classified as Amyloid positive (AMY+) or negative (AMY-). When applied on this “real-world” cohort, no statistical difference was found between ApV_1 and ApV_2 in subjects with positive/negative amyloid enhancement ($p > 0.05$) (Supplementary Figure 4A). Regardless of the PET output, subjects were classified as nAD_{rp} and MCI_{AD} (in particular, of the 44 AMY-, 42 were classified as nAD_{rp} , 2 as MCI_{AD} and 1 as AD; of the 38 AMY+, 36 were classified as nAD_{rp} and 2 as MCI_{AD}). The model was also tested on a subgroup of 22 subjects whose $T1w$ scan was obtained 5 ± 4 months after Amyloid PET imaging and was used together with the MMSE and the LDELTOTAL cognitive scores. In this small cohort, subjects with a negative PET scan were classified as nAD_{rp} ($n = 8$), MCI_{AD} ($n = 2$) and AD ($n = 1$). Subjects with a positive scan were evenly classified as nAD_{rp} and MCI_{AD} ($n = 5$), only 1 subject was classified as AD. In relation to the PET output, our ApV_{1s} showed a statistical difference between AMY- and AMY+ ($p < 0.05$) (Supplementary Figure 4B).

Lines 238 - 249

- Discussion: I believe this section would benefit of a deeper discussion of the selected features (which are never mentioned in the paper) and their possible meaning for AD, and a critical evaluation of the applicability of this method in real medical contexts.

Thank you for this comment. Our model extracted several statistical features which have been extensively described and related to biological phenomena in the file containing the Supplementary Material (Supplementary Note 1). Appreciating the importance that the interpretability of these features has in a future clinical translation, we modified the paragraph in the discussion:

The ApV describes the mesoscopic architecture and the biological changes of an AD brain. With an unsupervised approach, and appreciating the lack of post-mortem AD confirmation in our cohort of subjects, the algorithm selects texture and shape features, strong biomarkers of AD⁴⁴⁻⁴⁶, in regions typically involved in the development of the disease (the hippocampus, entorhinal cortex, amygdala⁴⁷). In particular, our results show a predominant dysfunction in the left hemisphere³³, confirming the strong left hemispheric lateralization found in the early stages of the disease compared to weak right hemispheric lateralization found in advanced stages³⁴. As extensively described in the “Biological interpretation of ApV” in the Supplementary Note 1, the cortical grey matter structural changes, usually due to ageing brain and cognitive decline caused by neuronal loss⁴⁸⁻⁵⁰, are represented in part within the ApV by GLCM and FD features⁵¹ and confirmed, with the multiparametric analysis of DTI MRI images, by the statistically significant decrease of FA in AD patients. For example, GLCM correlation feature, filtered with an LHL wavelet filter, in the left lateral ventricle expresses the dependency of grey level values to their respective voxels in the GLCM possibly relating to grey levels’ distribution in this brain region of AD patients where ventriculomegaly is commonly observed. Brain parenchymal shrinkage causes, in most neurodegenerative disorders, the passive enlargement of the lateral, third and fourth ventricles with a significant ventricular enlargement associated to AD⁵². Furthermore, cognitive decline, expressed as local neuronal loss of many hippocampal subfields (subiculum, cornu ammonis) following AD progression (as also confirmed by the statistically significant decrease of fractional anisotropy), is expressed by the Neighbourhood Grey Tone Difference Matrix (NGTDM) coarseness feature extracted in the right hippocampus. This is a measure of the average difference between the central voxel and its neighbourhood and is an indication of the spatial rate of change. A higher value indicates a lower spatial change rate and a locally more uniform texture. Together with high

pass wavelet filters applied on one dimension and a low pass one applied on the other two, the extraction of the coarseness in the hippocampus represents an index of heterogeneity. Interestingly, the algorithm also selects regions not commonly related to AD, such as the cerebellum and the ventral diencephalon. Together with a few studies reported in the literature^{53,54}, this outcome challenges the traditional view that white matter bundles in the cerebellum or in the ventral diencephalon are not affected by AD, possibly highlighting new therapeutic opportunities.

Lines 328 – 355

Minor Comments

- Please check the font used throughout the text as it is sometime changing
- Please check that abbreviations are always introduced at their first appearance in the manuscript
- Line 36: “for this reason” is actually connecting two clauses that are not causally linked (how does early amyloid relate to structural changes?)
- Line 173: It seems that there is a “with” in the wrong position, probably to be deleted Line 177
- Line 187: “applied unmodified to a different field strength”, it would probably better to be more specific and bring the reader attention back by specifying the 3T scanner. Line 189
- Line 226: “As before” seems unnecessary Line 199
- Line 312: “confirming results ...” seems to be better suited for a discussion section. *This has been deleted.*

Thank you. Those issues have been addressed.

Reviewer #3 (Remarks to the Author):

Review: Nature Communications in Medicine (New Journal)

Title: Discernment of the mesoscopic architecture of living Alzheimer’s disease brain aids disease detection

Summary: The authors developed a novel predictive model to classify patients with late-onset Alzheimer’s disease. They use their novel unsupervised method to derive an “Alzheimer’s Predictive Vector” that can distinguish between patients with Alzheimer related pathologies and without Alzheimer related pathologies. The authors claim that their method based on MRI imaging outperforms fluid biomarker measures and atrophy measures of the hippocampus.

A single-variant association study was performed to identify markers associated with the novel disease classifier. Markers in TOMM40 at the APOE locus were significantly associated with patients that were determined to show Alzheimer’s related pathologies.

While this manuscript describes an interesting approach to classify patients based on MRI scans and infers genetic associations using this novel phenotype, there is an overall lack of clarity concerning the experimental design and interpretation of the results. My major criticisms are listed below:

1.) The authors performed single-variant association analysis based on multiple novel phenotype which was inferred using their LASSO approach. They report statistical significance based on a WALD test, which is commonly used to extract p-values from a logistic regression model. It remains unclear what was compared in their genome-wide association study and how the analysis was performed. The authors also need to state what type of quality controls were performed on the data. The results report a genome-wide significant variant in TOMM40, which is at the APOE locus, the most significant risk factor for AD. This is to be expected when comparing AD cases with cognitively normal elderly controls. The authors need to clarify their statistical model for their GWAs analysis, provide a detailed overview of the variant association at the locus and adjust for APOE status when performing their single-variant association. Adjusting for APOE status, will allow the authors to determine whether this is a true independent association with variants in TOMM40 or an association signal purely driven by the

APOE4 allele and surrounding variants in LD.

Thank you for this comment. This has been addressed in the “Discussion” section where the following paragraph was added:

(...) TOM40 is located adjacent to APOE4, and the two genes are thought to be correlated with Alzheimer's disease due to linkage disequilibrium (Yu et al., 2007). Given that after adjusting for APOE4 allele status, rs2075650 is no longer significant, this suggests the TOM40 association signal is likely driven by the APOE4 allele and surrounding variants.

Lines 376-379

2.) Several previous studies have used ADNI imaging data to classify patients into distinct subgroups and predict conversion from MCI to AD. These studies reported high accuracy and similar prediction rates as reported in this manuscript. While several studies have used lasso or elastic net, others have used transfer learning methods for this task. These previous studies have provided proof of principle that this approach can be used to perform genetic mapping to identify markers associated with different imaging related phenotypes. The manuscript lacks a more detailed comparison to other methods, highlighting the novelty of their approach. The authors need to state in which cases their method outperforms existing methods in classifying AD patients.

Thank you for this comment. This is a very interesting point, and we added a small paragraph in the discussion as an added limitation to this study. A direct comparison with results from previously published work complicated as we structured the groups in a different way. Regardless, our ‘simple’ T1 only approach was as good or better than all other methods that objectively compared CN, MCI and AD groups (Supplementary Table 4). Our approach, compared to many other published models, aims to give a biological explanation of the features extracted and a better interpretability of the results (which will help clinical translation in the future, and which is hardly ever present in deep learning-based works). We compared our univariate method with the most commonly employed multivariate models and report the accuracy measurements in Supplementary Table 4:

	nADrp vs ADrp		MCI _{AD} vs AD	
	train	test	train	test
Ensemble Bagged Trees	87.5	31.1	66.9	11.2
Naïve Bayes	76.9	62.0	62.1	11.0
KNN	85.0	74.1	60.3	11.1
SVM	87.9	30.9	66.5	10.0
Our method	98.3	97.8	78.7	78.6

Furthermore, in the Discussion we added the following paragraph:

(...) Compared to the most common multivariate models present in the literature (Random Forest, Naïve Bayes, K-Nearest Neighbours and Support Vector Machine), our univariate analysis shows higher accuracy (Supplementary Table 4) and easier interpretability, thanks to the implementation of manually engineered features, facilitating clinical translation. (...)

Lines 286- 290

3.) The authors continuously overstate their findings. Language such as “state-of-the-art accuracy” (lines 332.f) or “high accuracy is remarkable” (lines 238.f) should be avoided. As noted in the previous point 2, the reported accuracy is comparable to previous studies (PMID: 27054198). In addition, the authors state that they used external data sets to confirm the results and that their method is

robust. However, it does not become clear how for example patients from the OASIS cohort were selected for their analysis and compared to the results from the ADNI data set.

Thank you for this comment. We avoided those type of language in the entire text. The OASIS cohort, together with 3T ADNI3 and IMC cohorts, have been used to test the model as “external datasets”. From the OASIS database, we selected healthy controls and AD patients, as classified by their consortium. In particular, we chose baseline scans, as we did for the 1.5T ADNI cohort used for training the model. This has been specified in the text. (Lines 124, 409).

4.) The current structure and organization of the manuscript makes it difficult to read. Figures and figure captions are placed throughout the manuscript, and it becomes difficult for the reader to distinguish between the text in the manuscript and the figure captions. Some of the plots included in the figures, such as the Q-Q plot or the Manhattan plot don't add additional value to the results section and should be moved to the supplements. Instead, a locus zoom plot highlighting the variant association at the APOE4 locus would be more helpful for the reader to interpret the results from the single variant association. Furthermore, a lot of important information is hidden in the supplementary notes (1+2) and at least the discussion related to the genetic association should be move to the main text. Also, supplement tables 1 and 2 either not formatted correctly or the pdf conversion from text did not work as intended.

Thank you for this comment. To ease the manuscript read, all the figures and tables have been moved to the end of the document. Furthermore, a locus zoom plot was included (Supplementary Figure 8) of genome-wide association study (GWAS) adjusted for Age and BMI.

REVIEWERS' COMMENTS:

Reviewer #1 (Remarks to the Author):

addressed the concerns from this reviewer. thanks

Reviewer #2 (Remarks to the Author):

no further comments

The authors did not make any additional revisions to the manuscript, as they followed the comments of the reviewers, who did not have anything to suggest.

REVIEWERS' COMMENTS:

Reviewer #1 (Remarks to the Author):

addressed the concerns from this reviewer. thanks

Reviewer #2 (Remarks to the Author):

no further comments